



# Weather and climate and their human impacts and responses during the Thirty Years' War in Central Europe

Rudolf Brázdil[1,2], Petr Dobrovolný[1,2], Christian Pfister[3], Katrin Kleemann[4], Kateřina Chromá[2], Péter Szabó[5,6], Piotr Olinski[7]

[1] Institute of Geography, Masaryk University, Brno, Czech Republic
       [2] Global Change Research Institute, Czech Academy of Sciences, Brno, Czech Republic
       [3] Oeschger Centre for Climatic Change Research, Bern, Switzerland
       [4] German Maritime Museum – Leibniz Institute for Maritime History, Bremerhaven, Germany
       [5] Institute of Botany, Czech Academy of Sciences, Průhonice, Czech Republic
[6] Department of Environmental Studies, Faculty of Social Studies, Masaryk University, Brno, Czech Republic
       [7] Institute of History and Archival Sciences, Climate Change Research Unit, University of Torun, Poland

*Correspondence to*: Rudolf Brázdil (brazdil@sci.muni.cz)

**Abstract.** The Thirty Years' War, which took place from 1618 to 1648 CE, was an armed military conflict in Europe. It resulted from the culmination of contradictions between advocates of the Roman Catholic and Protestant Churches during

the 17th-century reformation, as well as a power struggle for European political hegemony. This war brought about extensive devastation to Europe. Based on documentary evidence, this paper characterizes the climate, weather extremes, and economic and socio-political events in Central Europe during that time. Natural climate forcing indicates a gradual climate deterioration during the first half of the 17th century, associated with a decrease in solar activity towards the Maunder Minimum and increased volcanic activity. The mean temperatures in Central Europe from 1618 to 1648 were

significantly colder than the reference period of 1961 to 1990 in winter, autumn, and annually, while precipitation and drought means did not differ significantly from the reference. Summer temperatures, spring precipitation, and drought also exhibited significantly larger variability. Remarkably below-mean values, centered around the 1630s, characterized precipitation and drought fluctuations. These analyzed climatic patterns were accompanied by the occurrence of numerous weather extremes. Notably, late winter, late spring and early autumn frosts, floods, intense rain spells, and droughts affected

grain, fruit, and vine grape harvests, as well as the yields of other crops. These weather extremes contributed to various human impacts, such as food shortages (reflecting harvests and grain prices), famines, and epidemics. Ultimately, these events, along with the effects of the war, led to a decline in the population. The results obtained were discussed within the broader European context, taking into account climate, weather extremes, and socio-economic impacts.

## 1 Introduction

The relationship between armed conflicts and the natural environment has been a prominent focus of environmental historical research for several decades (Tucker, 2012). Recent studies have analyzed the environmental aspects of various



events, such as the two World Wars (Laakkonen et al., 2017; Tucker et al., 2018) and the American Civil War (Drake, 2015). Some works conceptualize the tactical usefulness of natural factors, while others examine the environmental consequences of wars (Smith, 2018). Weather and climate, although previously somewhat neglected, have increasingly gained recognition

in such studies (e.g., Hsiang and Burke, 2014; Noe, 2015). Regarding the 17th century specifically, Büntgen et al. (2011) highlighted the potential connection between temperature minima and settlement abandonment during the Thirty Years' War. Parker (2013) analyzed the fatal synergy effect of climatic extremes, food shortages, and war, which, in his perspective, led to the "global crisis" of the 17th century. In a more detailed case study, Degroot (2014) described how changing climatic patterns in the 1660s influenced the course of the Anglo-Dutch wars.

The development of historical climatology and paleoclimatology in Europe over the past two to three decades has led to the creation of numerous databases and papers dealing with detailed weather information, even with hourly-daily resolution (e.g., Brázdil et al., 2005, 2010; Glaser, 2008; Pfister et al., 2018; Pfister and Wanner, 2021). These resources have been subsequently used to develop long-term climate series and reconstructions (e.g., Luterbacher et al., 2004, 2016; Xoplaki et al., 2005; Pauling et al., 2006; Dobrovolný et al., 2010) with great spatial (local, regional, continental) and temporal

(monthly, seasonal, annual) resolutions. Even spatial distributions of main climatic variables based on gridded reconstructions (e.g., Luterbacher et al., 2004, 2016; Xoplaki et al., 2005; Pauling et al., 2006; Cook et al., 2015) as well as circulation patterns as maps (e.g., Luterbacher et al., 2002) are available. It opens new horizons for many interdisciplinary studies, including war events.

The Thirty Years' War (referred to as TYW) took place from 1618 to 1648 as an armed military conflict in Europe. It

emerged from contradictions between the Roman Catholic and Protestant Churches during the 17th-century reformation, as well as a power struggle for European political hegemony. This war brought about extensive devastation in Europe, resulting in significant loss of human lives, population decline, epidemics, famines, poverty, destruction of infrastructure, and abandonment of settlements. While the event and period have received considerable attention in historical research (e.g., Asch, 1997; Parker, 1997; Arndt, 2009; Wilson, 2009; Gotthard, 2016; Pike, 2023), climatological patterns during this time

have mostly been analyzed on millennial or centennial scales (e.g., Glaser, 2008; Parker, 2008, 2013; Pfister et al., 2018; Pfister and Wanner, 2021). Studies directly focused on the weather and climate of the TYW or its specific periods are less common. For instance, Lenke (1960) presented climatological data for 1621–1650 based on the observations of "Landgraf" Hermann IV from Hesse, Germany. Brázdil et al. (2004) analyzed meteorological records from Michel Stüeler of Krupka, northwestern Bohemia, covering the years 1629–1649. The patterns of the Little Ice Age were briefly discussed in relation to

the decade leading to the TYW (Duchhardt, 2017) and in the introduction to the TYW (Arndt, 2009). Stoffel et al. (2022) investigated the effects of a cluster of explosive volcanic eruptions in the 1630s and 1640s on climate and the related social-political context, including human responses in western and northern Europe, as well as China and Japan.

The present study uses high-resolution weather and climatic data extracted from rich documentary evidence and natural proxies to analyze fluctuations and variability of weather, climate, and their extremes during the TYW, 1618–1648, in



Central Europe (nowadays Germany, Poland, the Czech Republic, Slovakia, Switzerland, Austria, and Hungary). In the context of weather/climate variability, related human impacts and responses are analyzed.

## 2 The Thirty Years' War

In Europe, the first half of the 17th century was marked by political instability and warfare. The TYW primarily occurred within the Holy Roman Empire, driven by conflicts between different Christian denominations, including Catholics,

Lutherans, and Calvinists, as well as dynastic and territorial interests. However, it involved various European powers directly or indirectly throughout its duration (Brendle, 2010; Stoffel et al., 2022). The war consisted of several regional and European conflicts, comprising a total of 13 individual wars and 10 peace treaties (Kaiser, 2005; Dürr, 2010). It was unprecedented in terms of the prolonged duration and the large number of troops involved in combat. Additionally, almost all neighboring states of Germany were directly or indirectly engaged in the conflict (Repgen, 2015). The term "TYW" is not solely a

modern descriptor; even European contemporaries of the first half of the 17th century recognized that the ongoing succession of conflicts constituted a larger and more widespread war (Kaiser, 2005; Repgen, 2015).

The sense that conflict might be inevitable arose in the early 17th century. This was a time when many were burdened with eschatological concerns (Asch, 1997; Kaiser, 2005; Bähr, 2017). The increasing polarization among different Christian denominations disrupted the functioning of institutions within the Holy Roman Empire, including its Imperial Diet, which

served as a central forum for negotiations (Kaiser, 2005; Gotthard, 2010). Within the Imperial Diet, an important subset consisted of the Imperial Estates, which comprised both ecclesiastical and secular entities. These estates held the right to vote and govern in their respective territories, with the emperor being the only authority above them (Wilson, 2009; Parker, 2013).

The Prague defenestration on 23 May 1618, which involved members of the Bohemian Estates throwing three officials of the

Habsburg government out of a palace window, served as the event that ultimately triggered the war (Burkhardt, 2010; Münkler, 2017). This act of protest was directed against Ferdinand II, the newly appointed king of Bohemia (emperor from 1619 to 1637), who strongly advocated strict Catholic values and posed a threat to the religious freedom that had been granted to the Bohemian Estates by his predecessor, Rudolf II (emperor from 1576 to 1612), in 1609. In defiance, the Bohemian Estates proclaimed Frederick V of the Palatinate as their king (Schorn-Schütte, 2009). Despite their initial military

successes, the Bohemian Estates were eventually defeated by the Holy Roman Emperor, Ferdinand II, in the Battle of White Mountain on 8 November 1620. This marked the conclusion of the first phase of the TYW, known as the Bohemian War (1618–1620).

The subsequent phase of the TYW, known as the Palatinian War (1620–1623), unfolded within the Holy Roman Empire and was directed against Frederick V of the Palatinate (Kaiser, 2005; Rebitsch et al., 2019). In 1620, Spanish troops started

occupying portions of the Palatinate in opposition to the Calvinistic Frederick V, who subsequently lost his Prince-Electorship in 1623. Instead, the emperor granted this title to Maximilian I, Prince-elector of Bavaria, who was a fellow





Catholic. The pursuit of this prestigious title remained a significant objective for many belligerents until 1648 (Kaiser, 2005; Repgen, 2015). In 1625, the Danish king, Christian IV, joined the war, leading to the Danish-Lower Saxonian War (1625–1629). This conflict concluded in 1629 with victory for the emperor, resulting in his control over the northern part of the

Holy Roman Empire as well (Kaiser, 2005; Schorn-Schütte, 2009; Rebitsch et al., 2019).

In 1630, Gustavus Adolphus, the King of Sweden, arrived in Pomerania, marking the beginning of the Swedish War (1630–1635). The Swedish king could not accept the Habsburg's sphere of influence which now extended to the Baltic Sea (Kaiser, 2005; Rebitsch et al., 2019). Initially, the Swedes achieved victories, but they faced defeat against the emperor's troops in 1634. Taking advantage of the situation, the emperor utilized a blueprint for peace and drafted the Treaty of Prague in 1635,

aiming to strengthen his power (Kaiser, 2005; Repgen, 2015).

However, the balance of power underwent another shift in 1635 when France officially entered the war, marking a new continent-wide phase from 1635 to 1648 (Repgen, 2015; Rebitsch et al., 2019). Going against denominational lines, France, a Catholic power with hegemonic ambitions, joined the war in alliance with the Protestant belligerents. Their primary objective was to suppress Habsburg power on their borders and prevent any collaboration between the German and Spanish

branches of the Habsburg dynasty. The outcome of the war remained uncertain until France and the revitalized Swedish forces gained control and defeated Emperor Ferdinand III (emperor from 1637 to 1657). Faced with impending defeat in the mid-1640s, the emperor initiated peace negotiations (Kaiser, 2005; Münkler, 2017; Rebitsch et al., 2019).

After initial peace negotiations in Cologne and Hamburg, the proceedings moved to Münster and Osnabrück in Westphalia in 1644. This marked the first pan-European peace congress and served as a catalyst for the development of institutionalized

international diplomacy. Rather than negotiating directly, leaders sent 109 delegations representing 194 European rulers (Parker, 1997; Brendle, 2010; Repgen, 2015). The negotiations concluded on 24 October 1648, with the signing of the Treaty of Westphalia, which was subsequently ratified in Nuremberg in 1649/1650. It is worth noting that conflicts persisted in certain regions even after the treaty, such as the Franco-Spanish War (1635–1659), which ended in 1659 with the Treaty of the Pyrenees (Kaiser, 2005). However, the peace achieved was not long-lasting, as the following century and a half

witnessed numerous conflicts and further warfare (Völker-Rasor, 2010). During this period, war was considered a "natural condition between states" (Schorn-Schütte, 2009).

The Treaty of Westphalia played a crucial role in resolving significant religious and territorial issues for several European states. Within the Holy Roman Empire, the treaty functioned as a constitution, redefining the relationship between the emperor and the Imperial Estates (Brendle, 2010). However, many Imperial Estates, including some Catholic ones, did not

accept the emperor's authority as preeminent (Kaiser, 2005). The treaty recognized three Christian denominations: Catholics, Lutherans, and Calvinists (Schorn-Schütte, 2009). A return to the status quo was exemplified by the "normal year" of 1624 (Brendle, 2010; Dürr, 2010). This meant that the ruler's denomination in 1624 determined the denomination that the population should follow going forward. This safeguarded individuals from the ruler's religious conversions in the past or future. However, those who chose a different denomination were required to convert or migrate to other parts of the Empire,

Europe, or the New World (Schorn-Schütte, 2009; Völker-Rasor, 2010). As Parker (2013, p. 211) states, "*the loss and*



*displacement of people were proportionately greater than in the Second World War, the material and cultural devastation caused were almost as great; and both the catastrophe and its aftermath lasted far longer*".

## 3 Data

### 3.1 Documentary data

Central Europe is part of a region with a rich collection of documentary evidence related to weather, climate, and their impacts and responses (Brázdil et al., 2005, 2010). It is characterized by a wide variety of documentary sources, which are listed with selected examples below:

(i) **Narrative sources**

Narrative sources (annals, chronicles, memoirs, inscriptions) describe, in varying levels of detail, mainly important

weather/climatic anomalies and related phenomena that are significant from the point of view of human memory or their impacts on human society. The entire period of the TYW in Europe is covered by *Theatrum Europaeum*, which includes descriptions of damaging thunderstorms, hailstorms, windstorms, optical phenomena, and other occurrences related to the weather. For instance, a report from 1635 describes severe cold conditions as follows (Oraeus, 1644, pp. 414–415): "*In this year* [1635], *such terrible cold began around its onset and persisted that wolves froze* [...] *which seems unbelievable;*

*similarly, in Bohemia, many soldiers from three regiments froze to death while marching just two miles, with over 50 soldiers dying suddenly due to the frost*". Apart from these general European narrative sources, numerous regional sources exist that cover smaller areas and provide valuable information about the weather. One example is a note found in the diary of Michael Hancke regarding the winter of 1636/37 in Gdansk, Poland (for the location of places, refer to Fig. S1): "*This winter was marked by windstorms, snow, and brief, moderate freezes.*" (archival source AS2).

(ii) **Weather diaries**

Weather diaries typically consist of qualitative visual observations and descriptions of weather and related phenomena. In Hesse, Germany, Landgraf Hermann IV recorded daily weather observations in Kassel from 1621 to 1640 and in Rotenburg an der Fulda, located approximately 39 km away, from 1641 to 1650. For instance, on 3 May 1639, he noted (Lenke, 1960, p. 63/7): "*Morning: wet weather, nothing but rain and fog, at the same time warm, east* [wind]. *Noon: the same. Vespers:*

*big rain, east* [wind]. *Evening: nothing but rain during entire night, east* [wind]". Using Hermann's records, Lenke (1960) calculated monthly frequencies of days with different characteristics of temperature, precipitation, wind patterns, and meteorological phenomena, including their occurrence dates. In Fischingen, located in northeast Switzerland and situated at an altitude of 625 meters above sea level near the Hörnli, a pre-Alpine peak reaching 1100 meters, Placidus Brunschwiler, the abbot of the monastery, regularly included daily to seasonal weather observations in the monastery's diary between 1616

and 1654 (AS14). For example, on 27 April 1631, he noted: "*Fresh snow fell*". Brunschwiler's observations provide unique information on temperature and precipitation patterns in a mountainous area throughout the entire period of the TYW. In the Jesuit diary of Trnava, Slovakia, György Dobronoki published detailed weather observations for 1636−1637 (weather-



related data published in Réthly, 1962). For instance, on 26 January 1636, the weather was so mild that people worked outside the entire day in their shirts without overcoats.

(iii) **Letters**

Letters, whether of an official or private nature, can provide valuable information about weather and related phenomena. For instance, the noblewoman Zuzana Černínová wrote to her husband on 26 September 1642 from Radenín in Bohemia, mentioning the "*very cold nights and big frosts*" during sheep shearing (Dvorský, 1886, p. 51). Another example is a letter from Count Miklós Esterházy to his wife in 1630, where he reported the severe weather around Trenčín in Slovakia on 22–

23 December. He expressed relief at being without his family during that time, as he would not have known "*how to protect the little children from the cold*" (Merényi, 1900, p. 53).

(iv) **Weather compilations**

Several authors have compiled information on past weather and related phenomena from various sources and published them as weather compilations. Central Europe, for example, is covered by several volumes, such as Weikinn (1961) for Europe,

Réthly (1962) for the former Hungary (Hungary, Slovakia, Transylvania), and Girguś (2022) for Poland (though he relies on data from Weikinn, 1961 for the TYW). Additionally, Militzer (1998) prepared a collection of weather and environmental data for Germany from 1500 to 1800. When using data from such compilations for research purposes, it is essential to critically evaluate them to avoid potential errors, such as misdating or duplication of phenomena (Bell and Ogilvie, 1978).

(v) **Electronic databases**

Several research groups have developed electronic databases that contain documentary data on weather and related phenomena, with a significant amount of data covering Central Europe. One such database is German Tambora (Tambora, 2023), as described by Riemann et al. (2015). Another database, Euro-Climhist (Euro-Climhist, 2023), provides convenient access to hundreds of weather observations, mostly focused on Switzerland (Pfister et al., 2017). A lesser-known database is the Polish database titled "Klęski elementarne" (Polskie Towarzystwo Historyczne, 2016).

**3.2 Climatic reconstructions**

The analysis of climatic patterns in Central Europe during the period of 1618–1648 relies on the following quantitative climatic reconstructions:

(i) Annual and seasonal temperature reconstruction for Central Europe (Dobrovolný et al., 2010) based on temperature indices derived from documentary evidence for the Czech Lands, Germany, and Switzerland (1501–1854 CE), and measured

temperature series from 11 climatological stations in Austria, Czech Republic, Germany, and Switzerland (1760–2007 CE).

(ii) Gridded ($0.5° \times 0.5°$) mean seasonal temperatures (Luterbacher et al., 2004; Xoplaki et al., 2005) for the European land (25° W–40° E; 35–70° N), derived from temperature-sensitive natural proxies and documentary data (1500–1900 CE), along with instrumental measurements (1901–2002) from Mitchell and Jones (2005).

(iii) Series of seasonal and annual precipitation totals for the Czech Lands (Dobrovolný et al., 2015) (1501–2010 CE),

derived from documentary-based precipitation indices (1501–1854) and mean areal precipitation totals for the recent Czech



Republic (1804–2010 CE). Documentary-based precipitation indices in a 7-point monthly scale are also available for Germany (https://www.tambora.org) and Switzerland (https://www.euroclimhist.unibe.ch/).

(iv) Gridded (0.5° × 0.5°) mean seasonal precipitation totals (Pauling et al., 2006) for the European land (30° W–40° E; 30–71° N), based on long instrumental precipitation series, documentary-based precipitation indices, and natural proxies (1500–
1900), combined with a gridded reanalysis (1901–2000) from Mitchell and Jones (2005).

(v) Seasonal and annual self-calibrated (sc) Palmer Drought Severity Index (PDSI; Palmer, 1965) for the Czech Lands (1501–2015 CE) (Brázdil et al., 2016), derived from Central European temperature and Czech precipitation reconstructions (Dobrovolný et al., 2010, 2015).

(vi) Gridded (0.5° × 0.5°) summer scPDSI from The Old World Drought Atlas (OWDA), based on tree ring widths (0–2012
CE) (Cook et al., 2015).

### 3.3 Other data

In order to present climatic patterns of 1618–1648 in relation to external natural forcings, the following two series were used:
(i) Reconstructed annual values of total solar irradiance (Lean, 2018).

(ii) Volcano forcing reconstruction expressed by stratospheric volcanic aerosol optical depth (SAOD) in the 30–90°N
latitudinal band (Crowley and Unterman, 2013).

To characterize the reflection of grain harvest and the related political and socio-economic situation in the availability of food, the following series of grain prices were used:

(i) Wheat prices of Braunschweig and rye prices of Augsburg, Cologne, Nuremberg, Würzburg (all in Germany), and Gdansk from Esper et al. (2017).

(ii) Wheat prices of Dačice (Czech Lands) from Brázdil and Durďáková (2000).

### 4 Methods

Fluctuations in climatic patterns during the period of 1618–1648 are expressed as anomalies or percentages relative to the 1961–1990 reference period. This reference period was chosen over the more recent climatic normal of 1991–2020, which has already been significantly influenced by global warming, resulting in notable changes not only in temperatures but also
in other climatic variables (see Brázdil et al., 2022). Additionally, some of the European reconstructions used in the analysis extend back to earlier periods than 2020 (see Sect. 3.2).

To account for the considerable inter-annual variability in the climatic variables studied, year-to-year fluctuations are complemented with a low-pass Gaussian filter that highlights decadal variability. The same type of smoothing, using various time windows, is applied to emphasize long-term trends in external forcings and weather extremes. Climatic patterns are
presented for the seasonal (winter DJF, spring MAM, summer JJA, autumn SON) and annual series. The statistical



significance ($p < 0.05$) of differences between the TYW climate and the reference period in Central Europe is tested using Student's T-test for means and Fischer F-test for variances (Wilks, 2011).

European gridded temperature, precipitation and scPDSI reconstructions were used to show their spatial variability over Central Europe by creating corresponding maps. For the window delimited by 4–25° E and 45–56° N, means of all related

grids were calculated and together with minimum and maximum grid values they were indicated at the top of such maps.

Grain price time series from seven Central European cities were analyzed with respect to significant structural changes. They represent parts of the series that differ significantly from each other in the mean value of grain prices. The most probable years of these level changes are subsequently identified as breakpoints. This analysis was performed using the R-package strucchange (Zeileis et al., 2002).

## 5 Results

### 5.1 Climatic patterns in 1618–1648

### 5.1.1 Temperature

Fluctuations in the mean seasonal and annual temperature series of Central Europe during the period of 1618–1648 (Fig. 1) exhibit significant inter-annual variability, with predominantly negative anomalies compared to the 1961–1990 reference

period. The coldest extremes occurred in the 1620s, specifically in the years 1621 and 1628 for JJA, 1627 for MAM, and 1627 for the annual series. The coldest extreme in DJF occurred in 1635, and in SON it was in 1640. On the other hand, the warmest extremes were detected in 1625 for DJF, 1629 for the annual series, 1638 for MAM, 1645 for JJA, and 1647 for SON. Above-mean temperatures typically occurred in short intervals of 1–3 years, except for MAM from 1645 to 1648, the annual series from 1644 to 1647, and JJA from 1642 to 1646. Below-mean temperatures, on the other hand, persisted for

longer durations, particularly in DJF (1618–1624 and 1634–1642), as well as in SON (1632–1643).



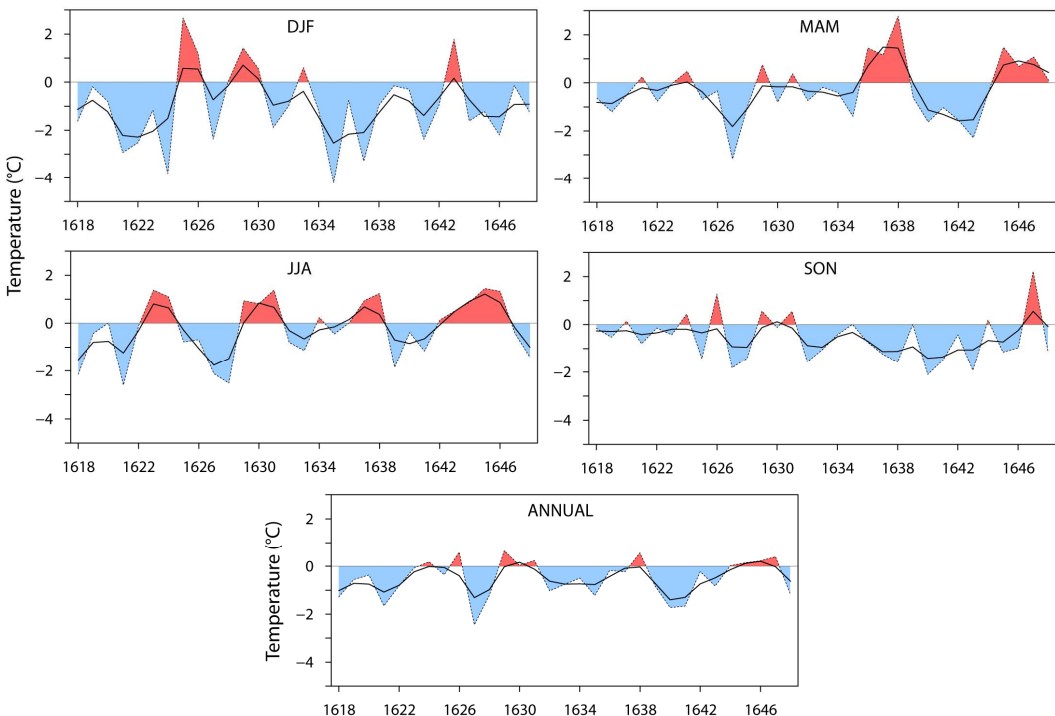

**Figure 1: Fluctuations in mean seasonal and annual temperatures in Central Europe during 1618–1648 expressed as anomalies with respect to the 1961–1990 reference (further as w.r.t. 1961–1990). Smoothed by 5-year Gaussian filter (data: Dobrovolný et al., 2010).**

According to the gridded European seasonal temperatures by Luterbacher et al. (2004) and Xoplaki et al. (2005), Central Europe as a whole experienced significantly cooler patterns ($p < 0.05$) than the 1961–1990 reference period in all seasons except JJA (Fig. 2). In DJF, the coldest area extended from eastern Germany towards the east and south, excluding the southeastern part of Central Europe. In MAM and SON, the coldest area formed a broad belt stretching from the Baltic Sea southwards across the central part of Central Europe, with an additional cold core over Switzerland in SON. In contrast,

warmer patterns were observed in JJA, particularly in the southeastern part of Central Europe.





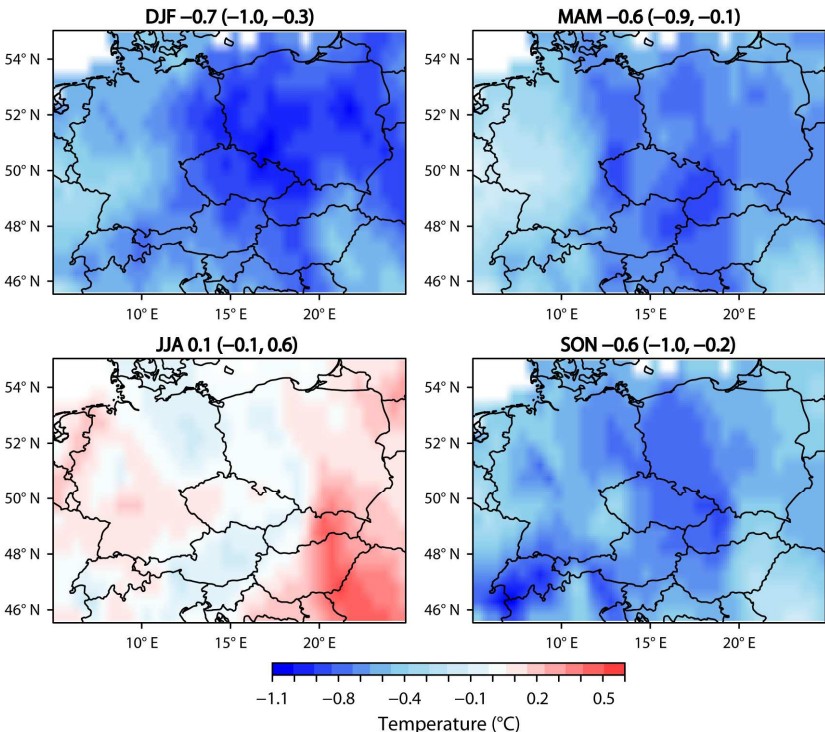

**Figure 2: Spatial distribution of mean seasonal temperatures in Central Europe during 1618–1648, expressed as anomalies w.r.t. 1961–1990 (data: Luterbacher et al., 2004; Xoplaki et al., 2005). The numbers at the top indicate mean (minimum, maximum) temperatures (°C) w.r.t. 1961–1990.**

**5.1.2 Precipitation**

Fluctuations in mean seasonal and annual precipitation totals over the Czech Lands during the 1618–1648 period (Fig. 3a) exhibit a high degree of inter-annual variability, characterized by sharp and short-lived increases and decreases. Notably, lower annual precipitation totals were observed from 1630 to 1634, with corresponding below-mean totals in DJF from 1631 to 1633, MAM from 1631 to 1634, and JJA and SON in 1630–1631 and 1634. However, below-mean totals persisted for

longer durations. Above-mean totals were typically limited to 1–3 years, although longer episodes with values close to or slightly below the mean were observed from 1624 to 1630 for DJF, 1636 to 1641 for JJA, and from 1644 to 1648 for SON. Similar fluctuations, as seen in the Czech series, can be observed in the documentary-based seasonal and annual precipitation indices for Germany (Fig. 3b). The series of Swiss precipitation indices (Fig. 3c), which have many missing or incomplete data, show less agreement with both the Czech and German series. This discrepancy may be attributed to problems with the



documentary data (missing monthly indices) and the different precipitation variability observed across various parts of Central Europe.

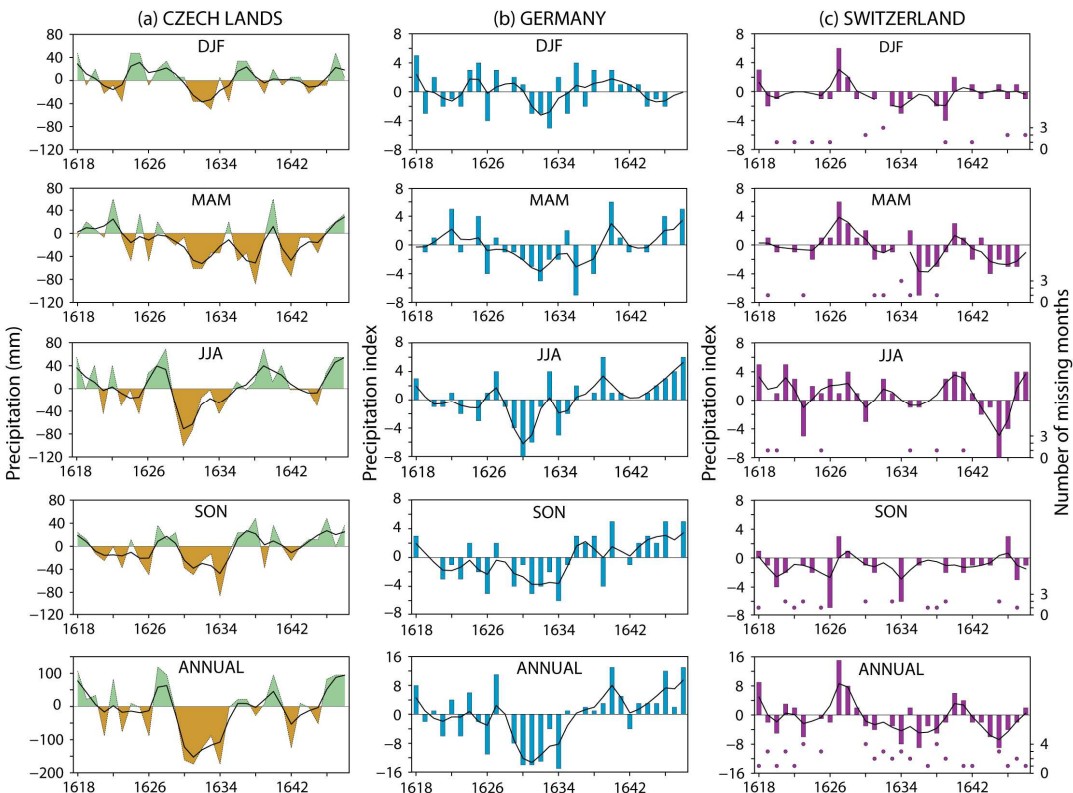

**Figure 3: Precipitation fluctuations in Central Europe during 1618–1648: (a) anomalies in mean seasonal and annual precipitation totals in the Czech Lands w.r.t. 1961–1990 (data: Dobrovolný et al., 2015); (b) documentary-based seasonal and annual precipitation indices for Germany (data: https://www.tambora.org); (c) same as (b), but for Switzerland (data: https://www.euroclimhist.unibe.ch/): points declare the number of missing months in calculation of seasonal or annual indices. Smoothed by a 5-year Gaussian filter.**

According to gridded European seasonal precipitation totals by Pauling et al. (2006), Central Europe experienced lower precipitation totals during the period of 1618–1648 compared to the reference period of 1961–1990 in DJF and SON, with

localized areas of both higher and lower percentages (Fig. 4). In MAM, the most pronounced precipitation decreases were observed in the northwestern and western parts of the region as well as in the greater part of Poland, while above-mean percentages were notable in a large area extending from eastern Switzerland to the northeast and east. In JJA, above-mean percentages were observed in a greater part of Central Europe with the highest values in eastern Austria and western




Hungary. Below-mean values occurred in several smaller regions. Central European JJA totals during the TYW were

significantly higher ($p < 0.05$) compared to the 1961–1990 reference period.

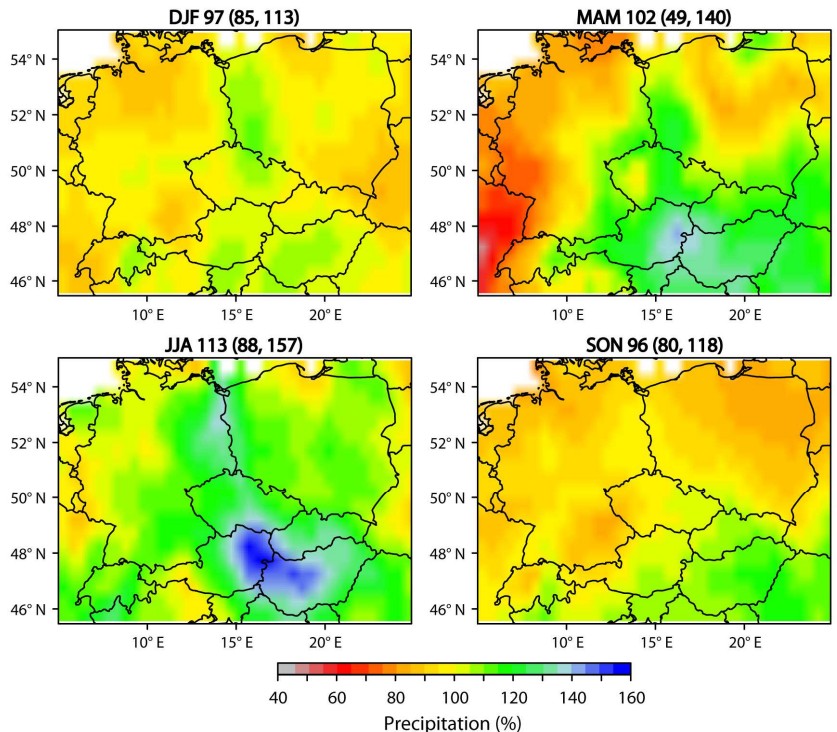

**Figure 4: Spatial distribution of mean seasonal precipitation totals in Central Europe during 1618–1648 expressed as percentages w.r.t. 1961–1990 (data: Pauling et al., 2006). The numbers at the top indicate mean (minimum, maximum) precipitation totals in % w.r.t. 1961–1990.**

**5.1.3 Droughts**

Drought patterns, represented by scPDSI in the Czech Lands (Fig. 5), demonstrate a notable and persistent dry anomaly in the 1630s, with the driest year recorded in 1632. This year is closely followed by the years 1631 and 1633. In contrast, wetter patterns (positive anomalies of scPDSI) were limited to shorter durations of 1–3 years. It is worth noting that scPDSI retains a "drought memory" meaning that drier and wetter episodes tend to be relatively consistent within individual seasons.



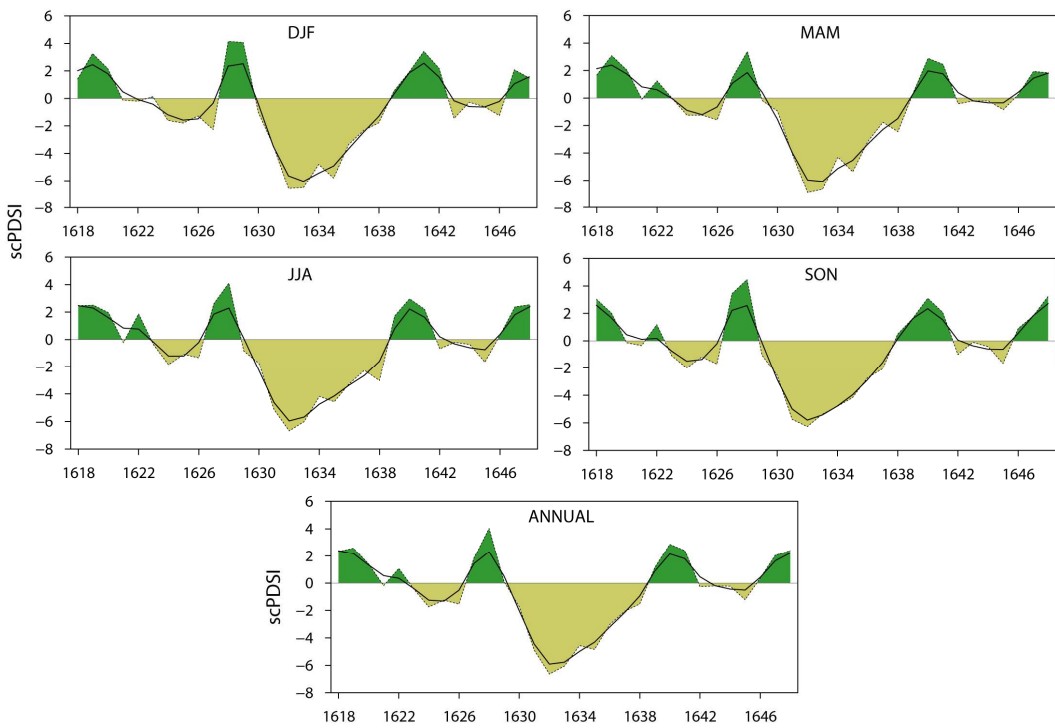


**Figure 5: Fluctuations in mean seasonal and annual scPDSI in the Czech Lands during 1618–1648 expressed as anomalies w.r.t. 1961–1990. Smoothed by a 5-year Gaussian filter (data: Brázdil et al., 2016).**

The spatial distribution of summer scPDSI, based on tree-rings (Cook et al., 2015), reveals complex patterns with prevalence of drier areas (Fig. 6). The driest area extends from northwest Germany to Bohemia, with intermittent wetter areas in a belt

stretching from northeast Germany to western Poland, the eastern parts of the Czech Lands, eastern Austria, and the western parts of Slovakia and Hungary (which partly aligns with the JJA precipitation totals in Fig. 4). The dry pattern continues in central and southeastern Poland. Additionally, a wet spot is observed in west-central Germany and in northeast Poland.

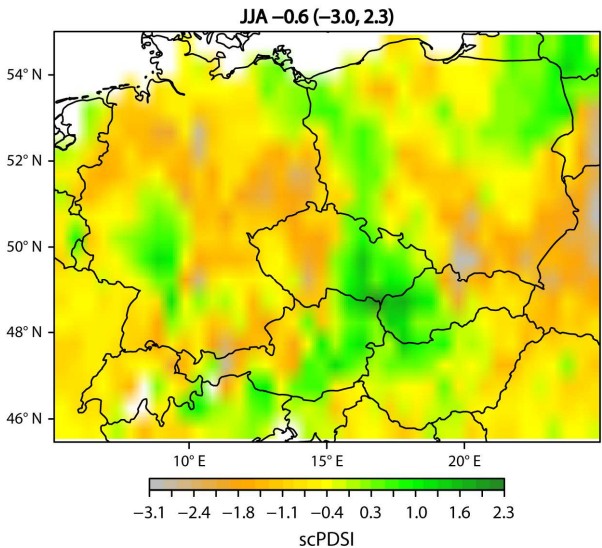

**Figure 6: Spatial distribution of summer scPDSI in Central Europe during 1618–1648 expressed as anomalies w.r.t. 1961–1990**
**(data: Cook et al., 2015). The numbers at the top indicate mean (minimum, maximum) scPDSI values w.r.t. 1961–1990.**

### 5.1.4 Long-term context and external forcings

To summarize the climate characteristics of the 1618–1648 period compared to the 1961–1990 reference, box-plots were created to show the seasonal and annual temperatures, precipitation, and scPDSI for both periods (Fig. 7). The results indicate that the TYW temperatures were significantly colder ($p < 0.05$) for DJF, SON, and annual values. However, there
were no significant differences in mean precipitation and drought values between the TYW and the reference period according to the T-test. The F-test revealed that the variances of JJA temperatures and MAM precipitation totals were significantly larger during the TYW. The scPDSI variability in all TYW seasons was also statistically significantly greater ($p < 0.05$) compared to the reference period, which is likely related to the reconstruction method used (Brázdil et al., 2016).





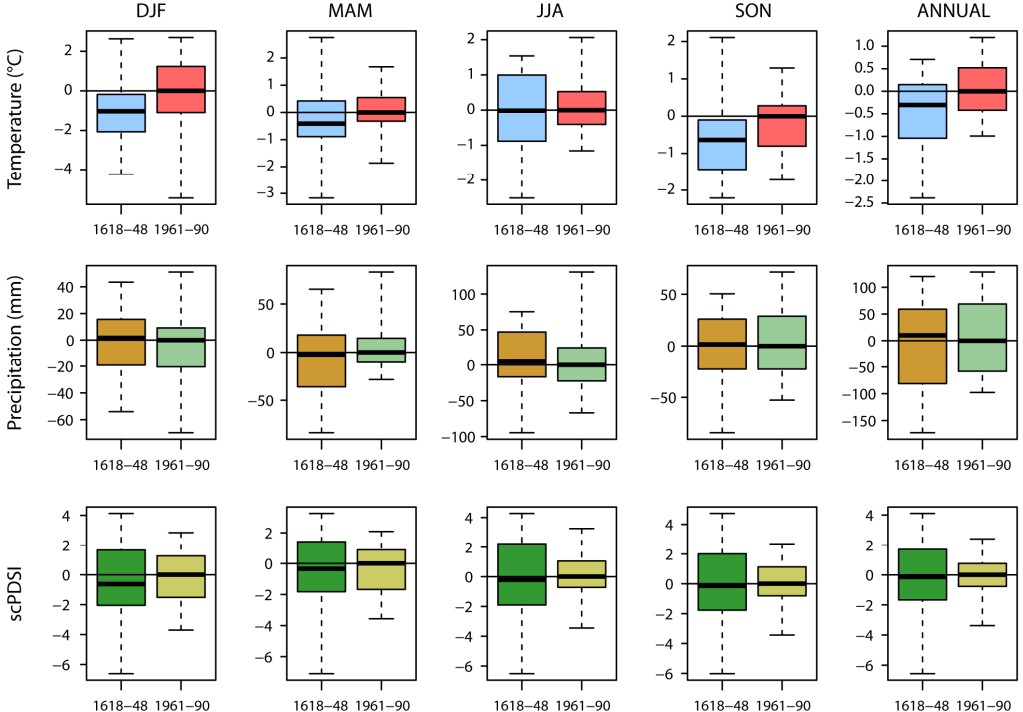

**Figure 7: Comparison of box-plots (median, upper and lower quartile, maximum and minimum) for mean seasonal and annual Central European temperatures (data: Dobrovolný et al., 2010), Czech precipitation totals (data: Dobrovolný et al., 2015) and scPDSI (data: Brázdil et al., 2016) for the 1618–1648 and 1961–1990 periods. All values are anomalies w.r.t. 1961–1990.**

In order to characterize the natural external forcings that contributed to the described climatic patterns, their fluctuations are depicted in Fig. 8. Regarding solar activity (Fig. 8a), there was a decreasing trend in annual total solar irradiance from the highest values at the beginning of the 17th century to the lowest values in the second part (Maunder Minimum, 1645–1715). Specifically, for the 1618–1648 period, the lower values observed until the early 1630s were replaced by a local maximum around 1640, followed by a steep decline. As for stratospheric aerosol optical depth (Fig. 8b), a clear volcanic signal emerged in the early 1640s, attributed to the eruptions of Komaga-take (Hokkaido, Japan, 31 July–9 October 1640, Volcanic Explosivity Index VEI = 5) and Mount Parker (Philippines, around 26 December 1640–4 January 1641, VEI = 5) (Global Volcanism Program, 2013). When considering the entire 17th century, a slightly stronger volcanic signal was only found in the mid-1690s.





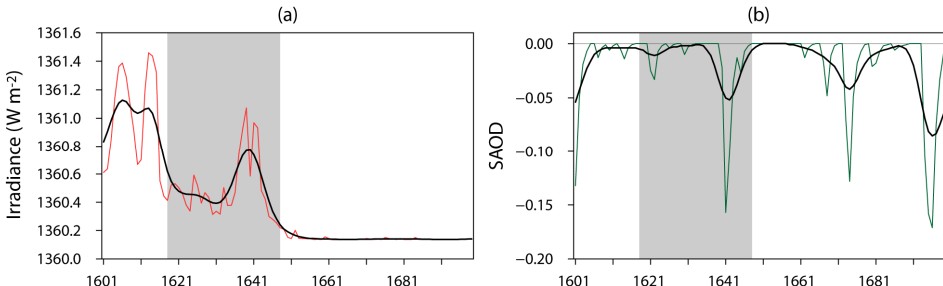

**Figure 8: Fluctuations in (a) total solar irradiance TSI (Wm⁻²; data: Lean, 2018) and (b) stratospheric volcanic aerosol optical depth SAOD (data: Crowley and Unterman, 2013) in 1601–1699 CE. Smoothed by a 15-year Gaussian filter. Gray band corresponds to 1618–1648.**

## 5.2 Weather and climatic extremes

Seasonal and annual numbers of frost, hot, and precipitation days, along with the dates of earliest and latest frosts and snowfalls, were derived from daily observations recorded by Landgraf Hermann IV in Kassel/Rotenburg an der Fulda for 1621–1648 (Lenke, 1960). These data were used to show fluctuations in weather and climatic extremes (Figs. S2 and 9). The highest annual number of frost days he recorded was in 1644 (102 days), followed by 1623 (87 days), and the lowest one in 1630 (40 days) followed by 1642 (43 days). The year 1644 also had the highest number of 49 hot days (followed by 44 in 1630) compared to only 5 hot days in 1636 and 12 such days in the preceding year, 1635. The highest number of 18 very hot days was recorded in 1638. The annual number of precipitation days fluctuated between 206 in 1648 and 123 in dry years 1631 and 1632. The latest MAM frosts occurred between 9 March 1630 and 28 May 1626 and 1639, and the earliest SON frosts between 9 September 1630 and 2 December 1633. For the case of the latest MAM snowfall, corresponding dates were between 19 March 1629 and 27 May 1639, and for the earliest snowfall, between 25 September 1643 and 22 December 1637.

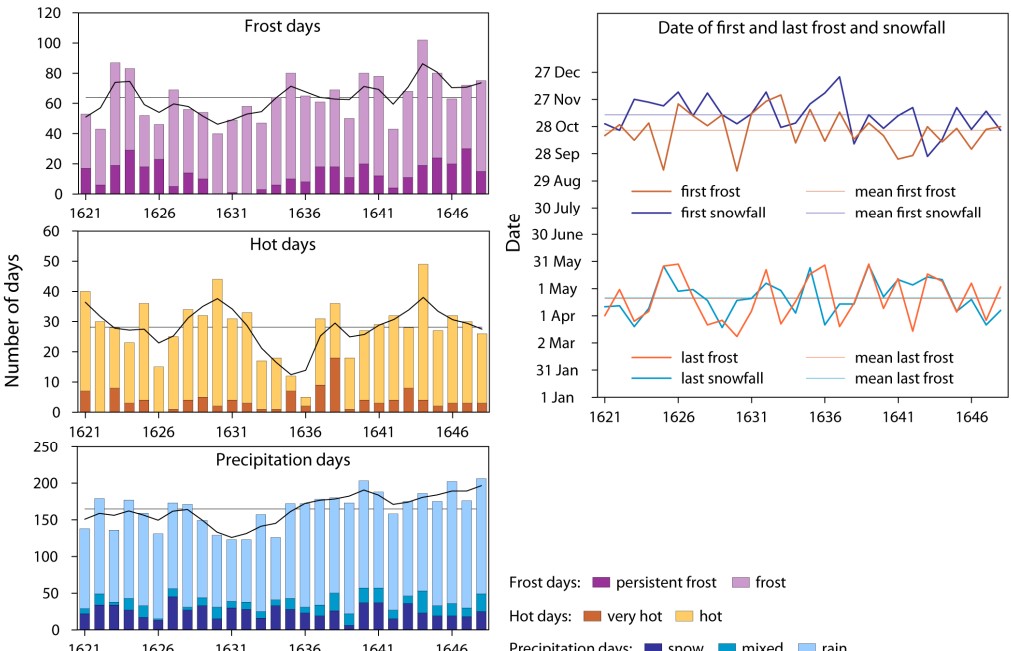

**Figure 9: Fluctuations in annual numbers of days with frost (frost, persistent frost), hot weather (hot, very hot) and precipitation (snow, mixed – snow and rain, rain) days and dates of the latest and earliest occurrences of frost and snowfall in Kassel/Rotenburg an der Fulda in the 1621–1648 period based on daily observations by Landgraf Hermann IV from Hesse (data: Lenke, 1960). Smoothed by Gaussian filter for 5-years. Corresponding means for 1621–1648 are indicated as horizontal lines.**

For the description of weather and climatic extremes and their adverse impacts on people and society in Central Europe during the years 1618–1648, there is extensive documentary evidence available (e.g., Glaser, 2008 for Germany). Due to the vast amount of reports, only a selection of them has been included in Sect. 5.3 or S1 in the Supplement. Table 1 provides a summary of selected extremes and their effects on different countries or regions based on detailed descriptions from each year. Several clusters of extreme events are notable, including late winter conditions continuing until March in 1619–1624; late frosts and snowfalls occurring during April–June in 1624–1629 and 1640–1645; floods during the summer half-year in 1618–1622; long dry spells during MAM and JJA in 1634–1638; long wet spells during MAM and JJA in 1625–1628; and early frosts and snowfalls in September–October in 1638–1644. The absence of certain regions in this overview may indicate either a lack of extreme events or the absence of documentary sources reporting such events.

**Table 1. Spatiotemporal summary of the occurrence of weather and climatic extremes in Central Europe during the 1618–1648 period according to S1 and Section 4.3. Extremes: LW – late winter, WF – winter half-year flood (October–March), LFS – late frost and snowfall (April–June), SF – summer half-year flood (April–September), DRY – drought (MAM, JJA), WET – wet (MAM, JJA), EFS – early frost and snowfall (September–October). Countries: A – Austria, CH – Switzerland, CZ – Czech Lands, D – Germany, HU – Hungary, PL – Poland, SK – Slovakia, Tr – Transylvania.**



| Year | LW | WF | LFS | SF | DRY | WET | EFS |
|------|------|------|------|------|------|------|------|
| 1618 | | | | CZ/D/Tr | | | |
| 1619 | CZ | CZ | CZ/D/HU | CZ | | | |
| 1620 | CZ | | CZ | CZ/SK | | A/CZ | CZ |
| 1621 | PL | | CZ | PL/SK | | PL | |
| 1622 | D | A | | CZ/D/PL | | CZ/D/PL | |
| 1623 | D | A | | | CH/CZ | A/D | |
| 1624 | CZ/D | | PL | | CH/CZ/D | | |
| 1625 | | | CZ/D/Tr | CZ | | A | A/CZ/D |
| 1626 | | | A/CZ/D | | | A/D/PL | |
| 1627 | CH/D | D | CH/CZ/D/SK | CZ/HU | | A | |
| 1628 | | | CZ | CZ/PL | | A/CH/CZ/HU/PL/SK | CZ |
| 1629 | | CZ/D | HU/SK | PL | | | |
| 1630 | PL | HU | | | CZ/D | | CZ/D |
| 1631 | | Tr | | | CZ/D/HU/Tr | | |
| 1632 | | | CZ/D/HU/SK | | | A | A |
| 1633 | | | CZ/HU/PL/Tr | | | | |
| 1634 | | | | | CZ/D | | |
| 1635 | D | | D/Tr | | D/SK | | |
| 1636 | | HU/Tr | | | D/CH/SK/Tr | | D |
| 1637 | D/PL/SK | D/SK | CZ/D | | CZ/PL/SK/Tr | | |
| 1638 | | | CZ/D | D | CZ/D/SK/Tr | | CZ/D |
| 1639 | | | | CZ/Tr | | A/CH/CZ/D/Tr | A |
| 1640 | | | CZ | CZ/SK | | D | A/CZ/HU/PL |
| 1641 | | | CZ/PL | | | A | A/CZ/D/Tr |
| 1642 | CZ | CZ | A/CZ/D/HU/SK | SK | D | | CZ/D |
| 1643 | | | A/CZ/D/PL/SK/Tr | D | | | A/CZ/D/SK |
| 1644 | D | | CZ/D/HU | D | PL | | CZ |
| 1645 | CZ | | CZ | | CH/HU | | |
| 1646 | | | | | A | | CZ |
| 1647 | PL | | CZ | HU/PL/Tr | | D | CZ |
| 1648 | PL | CZ/HU | | PL | | D | CZ |

### 5.3 1621, 1627 and 1628: three years without summer?

The Central European temperature series identifies (besides 1618) the years 1621, 1627, and 1628 as the three coldest summers (Fig. 1). These years, which are considered potential "years without summer", are further analyzed below.

In 1621, Central Europe experienced very cold weather with frozen wells and rivers in January and February in Moravia (AS5). Jakob Zetl reported great colds around 2 February with the frozen Enns River in Steyr, Austria (Klemm, 1983). The Untersee, a shallow part of Lake Constance, was covered in ice in January (AS8). Severe frosts in February in Silesia and



Saxony resulted in frozen rivers, pipes, and wells in cities and villages, leading to disrupted mill operations and causing hunger (AS10; Roch, 1687; Simon, 1696). In Saxony, frosts at the end of the month destroyed most of the sown grain,

leading to an increase in prices (Mörbitz, 1726). In the north of Poland, the cold and long winter lasted until the beginning of April (Czapliński, 2004). Frosts on 22 and 24 May in Bohemia caused damage to vineyards (AS11). Switzerland experienced cold rains in June; July and August had strong winds reminiscent of autumn months (AS14). In July, snow and strong frost were observed during the hay harvest on the Jura Mountains (Chaillet, 1845): "*You almost needed the sledge*". Many floods were reported in July and August in former Hungary, with one near Stará Lesná (now Slovakia) even washing

dead bodies out of their graves (Wagner, 1774). Poland experienced many rainy days during JJA (Zernecke, 1727) with the Oder flood in August (Weikinn, 1961). Cold and wet patterns during JJA in Central Europe are evident through negative temperature anomalies and above-mean precipitation compared to the 1961–1990 reference period (Fig. 10). The coldest areas were primarily located in the western half of Central Europe, while the wettest regions formed a belt along the Germany-Poland border and a cluster encompassing the southern Czech Lands, eastern Austria, Slovakia, and Hungary. The

Retz region experienced a poor summer with a smaller yield of sour wine (Lauscher, 1985). Following the first snowfall on 2 November, the weather in Louny in Bohemia (AS11) remained very cold until the end of the year (Čelakovský, 1875), similar to reports of frosts in Poland during SON (Weikinn, 1961; Czapliński, 2004). Various sources in the Czech Lands also mentioned scarcity or dearth during this year (e.g., AS11).

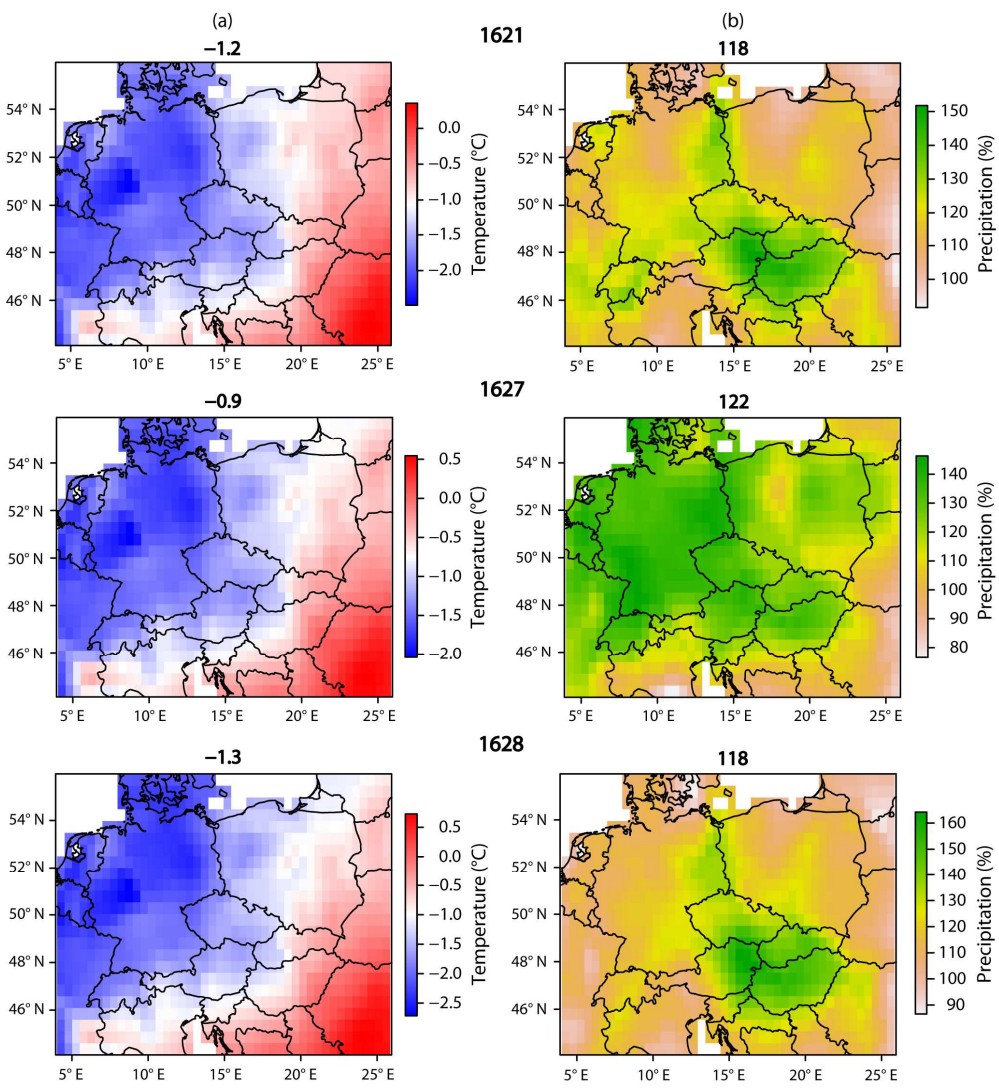

**Figure 10:** Spatial distribution of mean temperatures (a) and mean precipitation totals (b) in Central Europe during the summers 1621, 1627 and 1628 expressed as anomalies (temperatures) and percentage (precipitation) w.r.t. 1961–1990 (data: Luterbacher et al., 2004; Pauling et al., 2006). The numbers at the top indicate mean temperature anomalies (°C) and precipitation percentages (%) w.r.t. 1961–1990.



In the year 1627, MAM was exceptionally cold. Landgraf Hermann IV recorded 27 frost days in MAM in Kassel, including
11 frost days and 8 snowfall days in April (Lenke, 1960). Levoča in Slovakia also reported significant amounts of snow (Bal
et al., 1910). Switzerland experienced a permanent snow cover in March and April, with frequent snowfalls in May and even
some in June at altitudes of 600 meters above sea level (AS14). In the Ore Mountains (Czech-German border), snow and
frost occurred around 4 April and again around 22 June (Lehmann, 1747). Prague and Bohemia saw knee-deep snow on 29
April, which did not last long (Lisa, 2014). A flood on the Ohře River occurred on 9 June following heavy rain. Frost with
snow in Louny occurred on 21–22 and 24 June, with the cold being described as "*on glove*" (AS11). The JJA season in
Austrian Retz was characterized by miserable weather and frequent rains (Lauscher, 1985). In Hungarian Sopron, it rained
almost every day from late JJA onwards, resulting in two major floods and causing much of the harvest to rot and very sour
wine (Heimler, 1942). Cold and wet patterns during JJA in Central Europe are confirmed by generally negative temperature
anomalies and above-mean precipitation percentages compared to the 1961–1990 reference period (Fig. 10). The northern
half of Germany experienced particularly cold conditions, while the wettest area extended in a belt from Switzerland in a
northeastern direction towards the Baltic Sea and also in an eastern direction. September in Switzerland was cold and wet
(AS14), causing grape harvesting to be delayed until 5 November (Wetter and Pfister, 2013). In Austria and the Czech
Lands, the delay in grape harvesting was somewhat less pronounced (Maurer et al., 2009; Možný et al., 2016), while in Retz,
no wine ("Weinfehljahr") was reported (Lauscher, 1985). The Württemberg region in south Germany experienced multiple
river floods between 21 November and 2 January 1628 (Ginschopff, 1631). The entire year of 1627 in Bohemia was
characterized as cold and rainy, with average grain production and sour wine, unlike anything seen in the past 30 years
(AS11).

In 1628, Prague experienced a significant amount of snow on 24 January, which melted suddenly after five weeks (Menčík,
1897; Lisa, 2014). May witnessed an over-reproduction of cockchafers, causing damage in Bohemia (AS11; Lisa, 2014).
Between June 1st and 8th, it was very cold and windy, described as "*on cap and glove*" (AS11). Floods occurred on the Elbe
River on 10–11 and 22 July following prolonged rains (AS6). Continuous rain caused grain to rot in certain areas (Lisa,
2014), as well as in Levoča (Bal et al., 1910) and Sopron (Heimler, 1942). Poland also experienced a very wet JJA with
numerous rains and floods, particularly in central and northeastern parts and Silesia (Zernecke, 1711; Weikinn, 1961).
Switzerland witnessed cold and wet conditions from May to early August (AS14). Herdsmen reported 23 snowfalls below
2000 meters above sea level during the use of alpine pastures between late June and early August (Bärtschi, 1916). The
Swabian Mountains (around 1000 meters above sea level) were covered by a deep layer of fresh snow three times in July
(Militzer, 1998). The Württemberg region experienced wet and cold nights with strong winds in July, which affected the
harvest (Ginschopff, 1631). Several rivers, including the Neman and Vistula, flooded in West and East Prussia (Bock, 1782).
Retz reported cold and wet conditions during JJA (Lauscher, 1985). Processions for sunshine and warmth were even held in
Swiss Canton Valais, reflecting the dissatisfaction with the lack of sun and warmth (Bérody, 1889). A chronicler lamented
the sun's loss of its usual heat. Moreover, grain cultivated at higher elevations did not mature and there was no fruit
(Chaillet, 1845). Famine was reported in several areas due to frequent rainfall during the grain harvest, leading to sprouting



(AS9; Chaillet, 1845). Negative temperature anomalies compared to the 1961–1990 reference characterized JJA 1628 in Central Europe, with the northern half of Germany experiencing the coldest conditions. The highest above-mean precipitation were concentrated particularly in eastern Austria, southern Czech Lands, Slovakia and Hungary (Fig. 10). Grapes on the Swiss Plateau were picked on 15 November, which was even later than in 1816 (Wetter and Pfister, 2013). Similarly, there was some delay in French Beaune (Labbé et al., 2019) and in Bohemia (Možný et al., 2016), where vine grapes did not mature after frost on 19–21 September, resulting in sour wine (AS11; AS15; Mörbitz, 1726). In the Vienna region, the vintage started at its latest during the entire TYW, and the wine in the Retz region was of very poor quality ("Stößelwein") due to the very late harvest with unripe grapes, which had not been seen in 100 years (Lauscher, 1985). In Hungary, the grape harvest in 1628 was not particularly bad in terms of quantity, but it produced some of the worst-quality wines of the entire Little Ice Age (Rácz, 2020). Due to a small harvest of hops in Bohemia, their sale to foreigners was permitted. Mild weather prevailed there from 24 November to 5 January 1629 (AS11), as confirmed by only seven frost days in December in Kassel (Lenke, 1960), reports of mild November–December from other German sources (Simon, 1696; Militzer, 1998), a very mild December in Slavonia (Deák, 1867), and a very mild DJF in Poland (Zernecke, 1711).

### 5.4 Impacts and responses

#### 5.4.1 Weather extremes, war events, and population

The TYW left a trail of destruction in the lands of the Holy Roman Empire. Throughout the war, increased taxation put pressure on the economy (Ogilvie, 1997). None of the belligerents, except for the Dutch Republic, had enough cash and supplies to adequately support their troops. This was seen as a major factor in the undisciplined and cruel actions of the soldiers, particularly in areas where they passed through or camped (Asch, 1997). These actions took many forms: looting, burning down villages or cities, rape, mutilation, torture, and murder. The obligation to house troops, which fell upon the townspeople wherever the soldiers happened to be, furthermore led to the draining of local resources, scarcity of food, hunger, poverty, the risk of diseases spreading, and general hardship for all involved (Asch, 1997; Wilson, 2009; Münkler, 2017; Stoffel et al., 2022). The chronicler Hans Kriesche from Česká Lípa in Bohemia aptly expressed the misery of wartime (Panáček, 2016, p. 282, 284): "*Now I would like to close the old, bad, miserable, and sad year of 1620, in which there was nothing but unrest, war, murder, robbery, fire, hunger, and lack of water – a year that has never been forgotten in the history of the Czech Crown* […]".

Apart from the hardships of war, people also suffered from various weather extremes. For example, severe frosts affected both civilians and soldiers. In February 1624, many people froze to death in a forest at Hranice, Moravia, where they were hiding from Polish troops (Indra, 1940). Deadly losses to troops were reported around 22 December 1626, during the soldiers' return from Hungary to Moravia (AS3), as well as on 24 December during a campaign from Moravia to Silesia (Fialová, 1967). Many frozen soldiers were reported in Bohemia in 1635 (Oraeus, 1644). On the other hand, frosts could aid in aggressive actions. For instance, on 14 February 1621, around 2000 Hungarians crossed the frozen Morava River at





Brodské and looted the settlements of Podivín and Stará Břeclav (Zemek, 1968). This aligns with an earlier report by Gábor
        Bethlen that 2000 horses crossed the frozen Morava River (Moravia) in February 1621 (Szilágyi, 1879). Moreover, during
        the severe winters from 1644 to 1646 in the Ore Mountains, wolves were heard howling from hunger and killed dogs and
        cattle to survive (Lehmann, 1747).

        People on the German North Sea coast were particularly affected by storm surges. Three storm surges took place from late
January to March 1625, and on 20 February, the German Baltic coast was also impacted. Three more storm surges occurred
        there in May and from September to December 1628 (Happel, 1690; Petersen and Rohde, 1977). A very strong storm surge
        on 10/11 October 1634, known as the *Burchardi flood* or the second *Grote Mandränke*, affected the North Sea coast between
        North Frisia and Dithmarschen. It overran several dykes and washed much of the island of Strand away, drowning more than
        9000 people (Petersen and Rohde, 1977; Jakubowski-Tiessen, 2003; Allemeyer, 2009). On 24 February 1648, a storm surge
damaged churches and buildings, and toppled approximately 12,000 trees in Holstein, Hamburg, and Pinneberg (Schleder,
        1663; Happel, 1690; Petersen and Rohde, 1977). This storm also had an impact on Saxony and Brandenburg (Moller, 1653;
        Vogeln, 1714; Bekmann, 1751).

        Problems in everyday life were also connected to the occurrence of dry or wet periods in MAM and JJA. The lack of water
        caused difficulties not only for grain, but also for vegetables and fruit trees, and it often disrupted or halted the operation of
water mills. For instance, during the JJA of 1642, Leipzig in Saxony experienced a seven-week period without rain, and the
        dry weather persisted until November. This resulted in water scarcity, issues with operating water mills, and a poor fruit
        harvest (Weck, 1679; Vogeln, 1714). In 1631, a drought in Thuringia and Saxony caused wells and creeks to dry up, and
        forest fires broke out (AS7; Dreyhaupt, 1749). Water mills in Saxony were unable to function, leading to a shortage of flour.
        The drought also exacerbated food and fodder shortages (Abel, 1732; Mörbitz, 1726; Militzer, 1998). Conversely, periods of
heavy rainfall contributed to the occurrence of floods and complicated field work, particularly during harvest seasons.

### 5.4.2 Crop yields and prices

        The yields of agricultural crops, especially grain, were significantly influenced by weather conditions throughout their
        growth stages, including sowing, growth, maturity, and harvest. The available documentary evidence from Central Europe
        provides valuable information regarding the impact of weather patterns on grain production. These weather patterns affected
both the timing and the yields of grain harvests. During the TYW, the harvest dates of winter wheat in the Czech Lands were
        slightly delayed compared to the 1961–1990 reference period (Fig. 11a). On the other hand, the harvest dates of winter rye in
        Switzerland showed more variable cyclic fluctuations with the statistically significant downward trend ($p < 0.05$), indicating
        earlier harvest beginnings. This trend coincided with slightly increasing temperatures during the MAM and JJA seasons (cf.
        Fig. 1). However, there was some deviation in the early 1620s, disrupting the overall similarity between the two grain series.

Failures or poor harvests of grain were mainly associated with episodes of drought during the MAM and JJA seasons. For
        example, Pavel Mikšovic reported a severe drought and hot weather in Louny before 29 June 1631, resulting in dried-up
        brooks and wells, inoperable water mills, and frequent wood fires (AS11). These dry conditions led to a bad harvest,

particularly for spring grain, and caused dry grasslands, followed by shortages of fodder (AS12). Problems with grain also arose during prolonged periods of rain in the JJA season. In 1639, frequent rain spells during the JJA season in the Czech

Lands resulted in high yields of wheat and barley but poor-quality grain (Dvorský, 1886; Teplý, 1928). The wet JJA is confirmed by the recording of 61 precipitation days by Hermann IV in Kassel (Fig. S2c). In northeast Switzerland, there was mostly rainfall accompanied by cold winds from late May to 17 August, leading to delayed and wet hay and grain harvests (AS14).

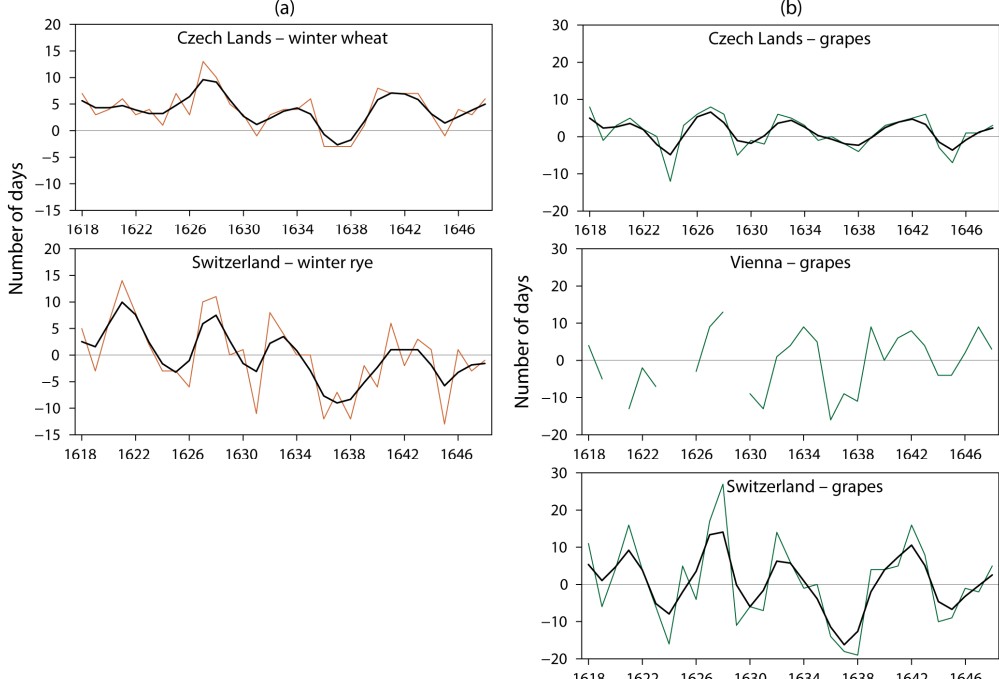

**Figure 11: (a) Anomalies in dates of winter wheat harvest in the Czech Lands (data: Možný et al., 2012), winter rye harvest in Switzerland (data: Wetter and Pfister, 2011) and (b) in dates of grape harvest in the Czech Lands (data: Možný et al., 2016), Vienna (data: Lauscher, 1985) and Switzerland (data: Wetter and Pfister, 2013) in the 1618–1648 period (reference periods: 1961–1990, for winter rye in Switzerland and grapes in Vienna 1618–1648). Smoothed by a 5-year Gaussian filter.**

Documentary sources indicate that the overpopulation of mice was a significant problem that caused damage to grain crops

during the TYW. Mice infestations affected various regions, including the Czech Lands, Silesia, and Brandenburg. In the Czech Lands, mice infestations were reported in 1622 (Fialová, 1967), 1623 (AS4; Fišer, 1931), and 1629 (AS1; Lisa, 2014), leading to damage to grain crops. In Silesia, a mice infestation occurred in 1624 (Militzer, 1998). The reports of mice infestations in Bohemia in 1637 (AS12) heralded their occurrence in Central Europe for the following four years. In 1638, mice were reported to have fed off the seeds in fields and caused destruction in barns in Bad Belzig, Brandenburg (Eilers,





1741). The Mulde River flood in Saxony in the same year led to the killing of many mice, which were previously a plague (Simon, 1696). The damage caused by mice feeding on grain and the passing of Swedish troops resulted in a significant increase in grain prices in Saxony, Thuringia, and Saxony-Anhalt in 1639 (Simon, 1696; Mörbitz, 1726; Dreyhaupt, 1749). In 1640, a shortage of grain and wine was reported as a result of mice consuming a large amount of the crops (AS12; Moller, 1653; Militzer, 1998). Mice also caused damage to sown grains during SON in Bohemia (Šůla, 1998) and Moravia (AS13;

Indra and Turek, 1946; Fialová, 1967). The mice population multiplied in fallow fields that could not be cultivated due to a lack of horses and people (Koch, 1914). In 1641, mouse infestations in Silesia led to crop damage, resulting in a poor harvest and price increases (Militzer, 1998).

The success or failure of grain harvests, along with war events and other socio-economic factors, had a direct impact on grain prices and subsequently led to food shortages and high prices. The highest grain prices occurred particularly in the

1630s and 1620s, as indicated by the prices of rye and wheat in selected towns (Table 2, Fig. 12). In the 1630s, the highest prices were recorded in Gdansk in 1630 and in south-German towns such as Augsburg, Nuremberg, and Würzburg in 1634. Dačice had its highest prices in 1639. Similarly, in the 1620s, Cologne experienced its highest prices in 1626 and Braunschweig in 1628. The six series with data for the entire period showed a significant increase in prices shortly after the beginning of the TYW, around 1620 or 1622. Another increase in prices occurred around 1650. The cheapest grain prices in

south-German towns were recorded either before or at the beginning of the TYW, while in Gdansk and Braunschweig, the lowest prices were observed in the mid-1650s. When examining detrended series, it can be observed that the prices before the TYW were at the same or higher level compared to the mean level after that period. Shortly after the outbreak of the TYW, there was a significant increase in grain prices in five of the analyzed series.

**Table 2. Years of the lowest (LP) and highest (HP) grain prices and breakdates in rye and wheat prices series of selected Central European towns in 1608–1658 (Dačice 1625–1658) (data: Brázdil and Durďáková, 2000; Esper et al., 2017).**

| Town | LP | HP | Breakdate |
|------|-----|-----|-----------|
| | | Rye | |
| Augsburg | 1618 | 1634 | 1620, 1627, 1632, 1637, 1645, 1650 |
| Cologne | 1647 | 1626 | 1612, 1622, 1643, 1648, 1653, |
| Gdansk | 1655 | 1630 | 1622, 1631, 1647, 1652 |
| Nuremberg | 1619 | 1634 | 1616, 1622, 1631, 1636, 1652 |
| Würzburg | 1608 | 1634 | 1620, 1626, 1632, 1637, 1642 |
| | | Wheat | |
| Braunschweig | 1656 | 1628 | 1622, 1630, 1635, 1642, 1648, 1653 |
| Dačice | 1633 | 1639 | 1631, 1635, 1639, 1646, 1650 |

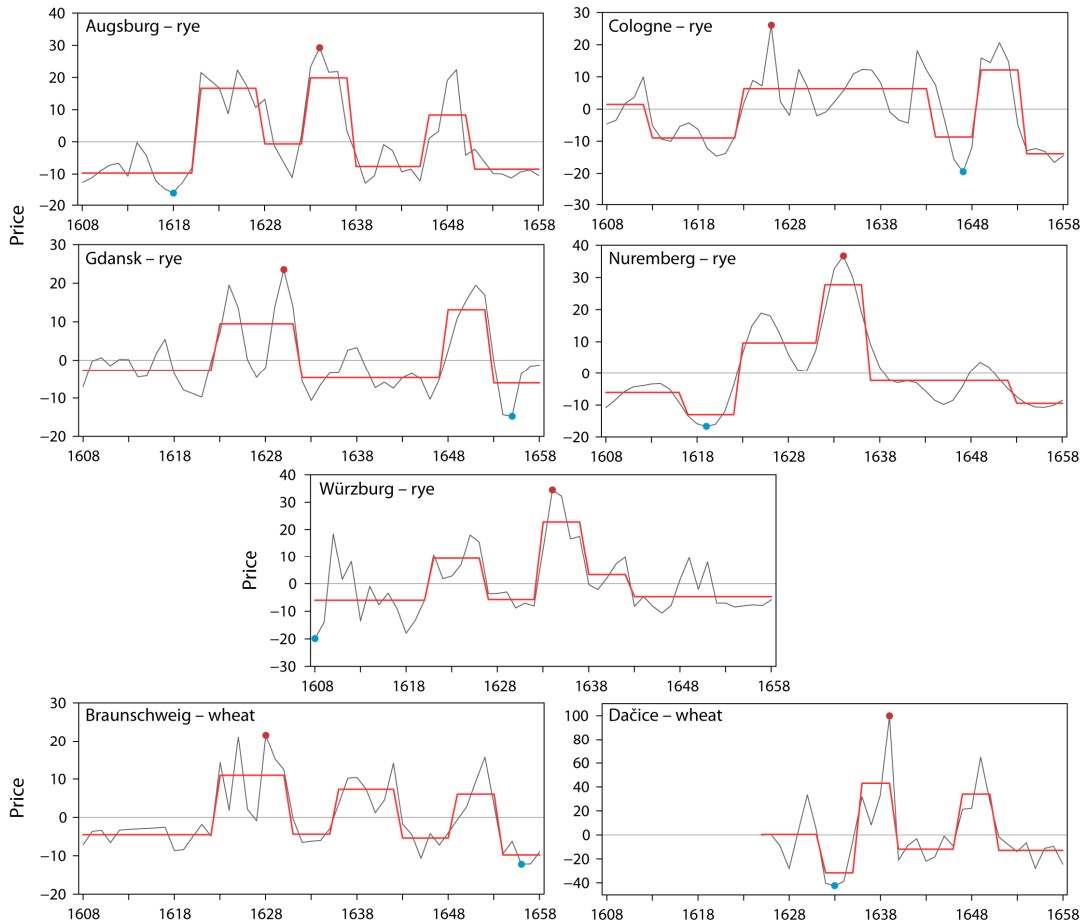

**Figure 12: Fluctuations in the prices of rye and wheat in selected Central European towns in 1608–1658 (Dačice 1625–1658) (data: Brázdil and Ďurďáková, 2000; Esper et al., 2017). Prices were detrended and expressed as anomalies from the mean of the whole series. Full red line shows variation of statistically significant changes (breakdates) in the level of each series. The lowest and highest prices are indicated by blue and red dots accordingly.**

In addition to grain, droughts also resulted in a shortage of straw and negatively impacted haymaking, which had negative consequences for livestock. In February and March 1637, György Dobronoki recorded a significant lack of hay and straw in Trnava, caused by the particularly dry period in JJA 1636 (Réthly, 1962). Late frosts, often accompanied by snow, occurring from April to June, and early frosts in September to October, could significantly affect the yield of vine grapes. The impact varied, ranging from no wine production to a very good yield. Temperature patterns from June to September also influenced





the dates of grape harvest, with consistent fluctuations observed in series from the Czech Lands (mainly Bohemia), the Vienna region, and Switzerland (Fig. 11b). Czech and Swiss series experienced non-significant downward trends, while the variation range of dates increased from north to south. Regarding wine quality, records from Retz indicated very good wine in four years, good wine in 11 years, average wine in three years, and bad (sour) wine or no wine in 13 years during the 1618–1648 period (Lauscher, 1985).

## 6 Discussion

### 6.1 Weather and climate

According to Pfister and Wanner (2021), the TYW did not experience over-freezing of Lake Zurich in Switzerland, despite the occurrence of severe DJFs in Central Europe, as depicted in Fig. 1. They argue that freezing events of lakes were more frequent during the broader period known as the Little Ice Age, but they were not observed during the TYW. Koslowski and Glaser (1999) developed an ice winter index based on accumulated areal ice volume along the German Baltic coast. They classified the winters of 1620/21, 1621/22, 1623/24, 1626/27, 1634/35, and 1645/46 as having a "very strong" ice winter index, which corresponds to the deeper temperature anomalies observed in Central Europe (Fig. 1). However, an "extremely strong" ice winter index was not recorded during the entire TYW period. Severe winters often resulted in a significant number of casualties due to frostbite (Jandot, 2017) as confirmed by more general reports for 1624 (Indra, 1940) and 1626 (Fialová, 1967) in the Czech Lands (Sect. 5.4.1) or other documentary data in S1. It is worth noting that long periods of extreme cold played a crucial role in warfare, as seen in later campaigns such as Napoleon's and Hitler's campaigns in Russia (e.g., Neumann and Flohn, 1987; Wiuff, 2023), not to mention similar conditions in earlier centuries.

Extreme summer temperature and precipitation patterns of 1621, 1627, and 1628 in Central Europe (see Fig. 10) indicate an association with "years without summer". According to the spatial reconstruction by Luterbacher et al. (2004), the distribution of temperatures in all three cold JJAs was more or less identical (cf. Fig. 10). It was characterized by negative temperature anomalies in the northwest and west of Central Europe, while there were positive temperature anomalies in the southeast. In contrast, the distribution of precipitation totals was spatially rather similar in 1621 and 1628 and very different in 1627. Overall, above-mean precipitation prevailed throughout the Central European area, indicating that the mentioned years were also significantly wetter compared to the reference climate. Moreover, for the years 1627 and 1628, this was confirmed by high scPDSI values reconstructed for Central Europe by Cook et al. (2015). Although the influence of the NAO on JJA temperatures in Central Europe is relatively weak, all three cold JJAs were characterized by a positive NAO index, resulting in enhanced zonal circulation (Luterbacher et al., 2001).

The spatial distribution of the lowest JJA temperatures in much of Western and Central Europe, growing towards the east, correspond to the typical conditions observed in the well-known year 1816 (Luterbacher and Pfister, 2015). When using the JJA Central European temperature series from 1501–2020 CE (Dobrovolný et al., 2010, extended), it is observed that JJA in 1816 was the second coldest, followed by 1621 (6th coldest), 1628 (7th coldest), and 1627 (17th coldest). The deepest





negative deviations during the 1618–1648 period were experienced in MAM and the year 1627, while the cold conditions in 1627 and 1628 were also well expressed in JJA and SON (cf. Fig. 1). Using a more recent JJA temperature reconstruction by

Luterbacher et al. (2016), although with less spatial resolution in the analyzed region, even greater cold anomalies (approximately 1.6 times larger) are observed compared to the findings by Luterbacher et al. (2004). The core of the cold anomalies is shifted towards southwest Central Europe and further west in this reconstruction.

Písek and Brázdil (2006) highlighted the strong influence of volcanic eruptions in Iceland or Italy on temperatures in Central Europe, particularly demonstrated after the Lakagígar eruption (VEI = 4) in June 1783 (Brázdil et al., 2017; Kleemann,

2022, 2023). The eruption of the Icelandic volcano Katla between 2 and 25 September 1625 (Frímann, 2011; Global Volcanism Program, 2013) with VEI = 5 was the closest in time to the years 1627–1628. While documentary sources reported May frosts in 1626 (S1), only 5 hot days and 49 precipitation days were recorded by Landgraf Hermann IV in Kassel during the following JJA (Fig. S2c), indicating a rather cold season. Similar patterns were recorded in the summer of 1636, with only 4 hot days and 52 precipitation days in Kassel, coinciding with the eruption of Hecla (VEI = 4) that started

on 8 May and lasted for about a year (Frímann, 2011). Additional Icelandic eruptions from the Grímsfjall volcano were reported on 29 July 1619, 1629, and around 24 February 1638 (all with a VEI = 2), as well as in 1637–1638 in the vicinity of the Vestmannaeyjar volcano (ibid.). The potential effect of the Grímsfjall volcano on the very cold JJA of 1621 in Central Europe (cf. Fig. 10) is less probable, despite reports of sulfur raining down on 30 May in many places in Germany (Simon, 1696; Militzer, 1998).

The strongly expressed volcanic signal in the early 1640s (cf. Fig. 8b), which was analyzed by Stoffel et al. (2022), is also evident in the Central European temperature series (Dobrovolný et al., 2010). There were significant negative anomalies in annual and SON temperatures in 1640–1641, and slightly less pronounced negative anomalies in MAM (persisting until 1643) and JJA (cf. Fig. 1). In this context, it is interesting to note the regular occurrence of late April–May frosts in the Czech Lands between 1640 and 1645, which even extended into JJA in 1641–1642 (see Table 1 and S1).

The various weather extremes reported for 1618–1648 in Table 1 and S1 can be compared with the results of other papers. For example, Blöschl et al. (2020) found two flood-rich periods in Europe during the time of the TYW, based on data from 103 flood series spanning from 1500 to 2016 CE. These periods were identified as 1590–1640 for Iberia and southern France (ranked 6 on a scale from 1 as the weakest to 10 as the strongest period) and 1630–1660 for western Europe, west-central Europe, and northern Italy (ranked 7). On the other hand, Glaser et al. (2010) identified 1640–1700 CE as a period of

increased flooding for 19 European rivers, with 12 of them located in Central Europe. The Central European documentary evidence in Table 1 identifies floods occurring in this region in at least 22 years (1618–1623, 1625, 1627–1631, 1636–1640, 1642–1644, and 1647–1648). Regarding significant droughts in the Czech Lands during the 1630s (cf. Fig. 5), it is well confirmed by outstanding drought periods during March 1630–May 1633 and March 1634–April 1635 in Germany (Glaser and Kahle, 2020). It also partially coincides with the 1626–1635 megadrought in the middle Ebro valley in Spain (Cuadrat et

al., 2022). This megadrought was characterized by a progressive loss of crop yields, food shortages, and malnutrition of the population, with an extraordinary peak of mortality in 1631.





### 6.2 Human impacts and responses

The time of the TYW is often associated with the so-called General Crisis of the 17th century: "*a combination of economic, social, climatic, political, and intellectual changes which made confrontations* [...] *far more likely*" (Parker, 1997, pp. 185–186). Other wars also took place in Europe during this period (Parker, 2013; van Nimwegen, 2014).

During the TYW, some European regions experienced significant population losses, ranging from 10 to 50 %, and in some cases even exceeding 70 % (Asch, 1997; Kaiser, 2005). Estimating the exact population of the Holy Roman Empire during that time is challenging, but it is believed to have been around 16 to 18 million in approximately 1618, and decreased to 10 to 12 million by around 1650 (Wilson, 2009; Repgen, 2015; Gotthard, 2016). This corresponds to a population decrease of 25 %. Some sources suggest an even higher decrease of 35 to 40 % (Asch, 1997). Regions such as Mecklenburg, Pomerania, the Palatinate, and parts of Thuringia and Württemberg, which were heavily affected by the war, experienced population losses of over 50 %, and in some places, over 70 %. In contrast, the northwest and southeast regions of the empire were relatively spared from depopulation (Dreißigjähriger Krieg, 2023). Pfister and Wanner (2021) mentioned a dramatic population decrease in Germany after the TYW, reaching levels similar to those in 1520. They attributed this decline to the combined effects of warfare, cold spells in the late 1620s and early 1640s, and waves of plague and dysentery. In the Czech Lands, depopulation is estimated to have been around 30 %, although the accuracy of this figure is questionable due to unreliable population estimates before the TYW (Maur, 1987; Fialová et al., 1996). However, a significant decrease in population is evident in the region through a substantial decline in building activity, which dropped by more than 30 % during the TYW compared to the previous 30 years (Kolář et al., 2022).

Weather and climate impacts on population size should be investigated in a broader context that also includes the effect of war and epidemics (Fig. 13). Major wars like the TYW devastated the livelihoods of the concerned populations. They overlapped weather-related supply crises and promoted the spread of epidemics, mainly the plague (Pfister and Wanner, 2021). Before the late 17th century, large-scale weather-related losses in food production were among the most important causes of subsistence crises, affecting population size in a largely self-sufficient, organic economy. Several points have to be stressed here:

(i) Population size and agricultural production have a positive feedback. More hands were needed to increase production, and in turn, increased production allowed for feeding a growing population.

(ii) Weather and climate primarily affected the production of biomass (food and feed).

(iii) Epidemics often had far-reaching demographic and economic consequences.

(iv) Major wars decimated the civilian population and destroyed their livelihoods.

(v) After the late 17th century, innovations such as the increase in agricultural productivity and improvement in transportation gradually reduced vulnerability to weather and climate.





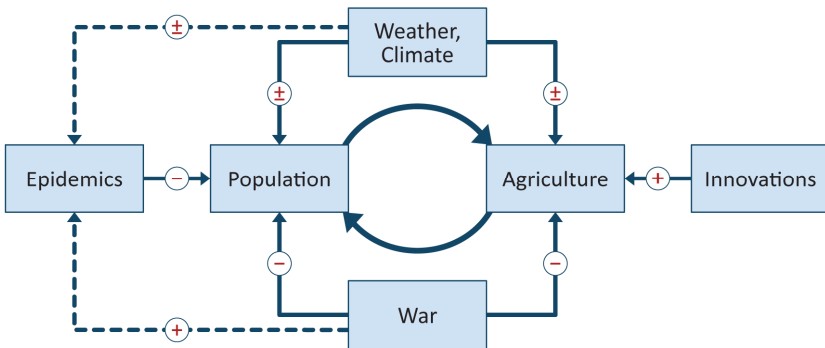

**Figure 13: A simplified scheme of four main external determinants affecting population size in a largely self-sufficient, organic economy. The plus symbol "+" indicates an enhancement of relationship and the minus symbol "–" its weakening (© Pfister/Wanner, *Climate and Society in Europe. The Last Thousand Years*. Haupt Verlag, 2021, p. 251).**

The war shifted the balance between the determinants in Fig. 13. Subsistence crises and famines were primarily caused by the war and its effects, devastating farmland, decimating livestock, burdening subjects with war taxes and tributes, and resulting in a significant loss of the workforce (Outram, 2001). Population losses were largely attributed to epidemics, especially the plague, which spread through the marauding soldiers and refugees seeking refuge in towns (de Waal, 1990). For instance, a plague was cited in 1622 in former Hungary (Réthly, 1962), in 1622 and 1646 in Velké Meziříčí in Moravia (Štindl, 2004), in 1632, 1635 and 1636 in Austria (Lauscher, 1965), etc. In such circumstances, the meteorological impacts only partially overlapped with the effects of the wars.

Clues about potential climatic impacts under these circumstances can be obtained by examining their meteorological causes rather than their potential human consequences. When considering the vulnerability of the main sources of food in the past, a prolonged lack of warmth coupled with excessive precipitation throughout the agricultural year was likely to affect all food sources. Pfister (2007) proposed a model of climatic impact factors leading to subsistence crises, which allows for an assessment of the biophysical vulnerability of food production, including grain, vine, and dairy products, over time. Such combinations of adverse factors resulting in overall crop failures were referred to as Little Ice Age-Type Impacts (LIATIMP).

The level of impact factors is interpreted as an approximate measure of the intensity of climatic stress. When examining the fluctuations of grain prices in Nuremberg between 1500 and 1670 CE (Fig. 14), it can be observed that the impact levels were low during the final phase of a relatively favorable climate period up until 1567. The transition to the Little Ice Age climate between 1569 and 1573 brought about a major peak of climatic stress and severe famine. Another peak is notable between 1626 and 1628. Upon closer examination of the data (refer to Sect. 5.3), it suggests that the inflation related to the war was amplified in 1627 and 1628 by the impact of a climate-related general crop failure. After 1630, despite the ongoing





war, the level of climatic stress decreased. Overall, the impact levels from 1568 to 1630 were significantly (p < 0.05) above the long-term mean (Pfister, 2007).

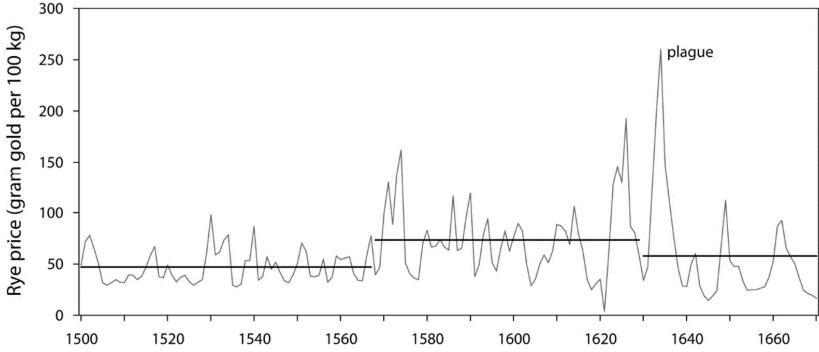

**Figure 14: Detrended market prices for rye (gram gold/100 kg) in Nuremberg, harvest years 1500–1670 (changed according to Pfister, 2007; data: Bauernfeind, 1993).**

According to the description of the course of the TYW in Sect. 2, different parts of Central Europe may have been affected differently. Germany and the Czech Lands were heavily impacted by war events, while the Swiss Confederation remained neutral. The former Hungary, on the other hand, was less affected compared to other parts of Central Europe. In terms of agricultural products, grain and hay harvests were generally poorer than usual during the TYW. However, the quantity and quality of grape harvests tended to be above average, especially in the Transdanubian region, which was geographically closer to Moravia and Austria (Rácz, 2020).

## 7 Conclusions

The analysis of weather and climate during the TYW in Central Europe from 1618 to 1648 led to the following conclusions:

(i) The temperatures during the TYW were significantly colder compared to the reference period of 1961–1990 for DJF, SON, and annual series. However, there were no significant differences in mean precipitation totals and PDSI compared to the reference period. The JJA temperatures and MAM precipitation totals showed significantly higher variability, as well as the PDSI values. The patterns of precipitation and drought during the TYW indicated a particular period of below-mean values in the 1630s. The spatial distribution of three climatic variables showed both widespread anomalies and localized variations in different regions.

(ii) The analysis of external forcing during the first half of the 17th century revealed a gradual deterioration of the climate. This was evident through a significant decrease in solar activity, with a temporary local increase around 1640, leading up to the period known as the Maunder Minimum (1645–1715). Additionally, there was an increase in volcanic activity, especially

during the early 1640s, which coincided with the highest SAOD signal. Prior to this, there were several Icelandic eruptions
between 1619 and 1638 that also contributed to the changing climate conditions.

(iii) The people during the TYW suffered not only from war hardships but also from weather extremes, such as severe winter frosts, late spring and early autumn frosts, storm surges, floods, droughts, windstorms, hailstorms, or torrential rain. These extreme weather events caused damage and even loss of human lives, as documented by rich Central European documentary evidence. While some of these events were localized or sub-regional, others affected larger areas. The weather extremes had
685 a significant impact, particularly on crop production. Regions that were less affected by local war events and troop movements were better able to cope with the negative effects of these weather events.

(iv) The importance of warlike actions during the TYW cannot be overestimated, as their long-term effects went far beyond weather-related crop failures. The unbridled soldiery, which had to sustain itself, burned down dwellings, barns, and stables, resulting in the decimation of livestock, which was essential as draught power and a source of fertilizer. Moreover, the war
caused a loss of the labor force and the regenerative capacity of the population, not primarily through military action, but through the spread of epidemics, especially dysentery and the plague. The large armies, numbering in the thousands, significantly increased the demand for grain. It is important to investigate the significance of climatic variations and extreme events using the example of territories that were spared from war, although it should be noted that epidemics did not stop at territorial borders. This study serves as a starting point for such investigations.

**Data availability.** The datasets and series used in this article are either publicly available (see citations of climate reconstructions) or can be obtained by personal request.

**Author contributions.** RB contributed with Czech documentary data and designed and wrote the paper with contributions
from all co-authors. PD dealt with the analysis of climatic patterns and forcings during TYW. CP contributed with Swiss data and discussion of human impacts and responses. KK prepared a detail description of TYW events and German documentary data. KC worked with the observations of Hermann IV and finalized all figures. PS contributed with documentary data from former Hungary and PO with Polish documentary data. All authors have read and commented on the latest version of the paper.

**Competing interests.** The contact author has declared that none of the authors has any competing interests.

**Special issue statement.** This article is not a part of the special issue. It is not associated with a conference.

**Acknowledgements.** RB and PD were supported by the Ministry of Education, Youth and Sports of the Czech Republic for SustES – Adaptation strategies for sustainable ecosystem services and food security under adverse environmental conditions project, ref. CZ.02.1.01/0.0/0.0/16_019/000079. KC was supported by the Global Change Research Institute of the Czech





Academy of Sciences, PS by the long-term research development project no. RVO 67985939 and PO by grants funded by the National Science Centre, Poland (Grants No. 2020/37/B/ST10/00710). We would like to thank Jan Esper (Mainz) for grain price series of six Central European towns and Laughton Chandler (Charleston, SC) for English style corrections.

**Financial support.** This research has been supported by the Ministry of Education, Youth and Sports of the Czech Republic (grant no. CZ.02.1.01/0.0/0.0/16_019/0000797), the Global Change Research Institute of the Czech Academy of Sciences, long-term research development project no. RVO 67985939 and the National Science Centre, Poland (Grants No. 2020/37/B/ST10/00710).

### Archival sources

AS1: Archiv města Ústí nad Labem, fond Sbírka rukopisů: Letopisecké záznamy Jana Čeledínka z Čáslavi připsané k Veleslavínovu Kalendáři historickému z r. 1590.

AS2: Biblioteka Polskiej Akademii Nauk w Gdańsku, MS 915: Sammelbuch des Danziger Schreibers Michael Hancke.

AS3: Moravský zemský archiv Brno, fond B 10 Komerční konses 1751–1775, inv. č. 1106: Zpráva o Švédovi, když přišel do města Olomouce. Kronika protestantského olomouckého měšťana z let 1620–1670.

AS4: Moravský zemský archiv Brno, fond G 10 Sbírka rukopisů Zemského archivu 1200–1999, inv. č. 1211/2: Opisy korespondence T. Pešiny z Čechorodu 1667–1680.

AS5: Moravský zemský archiv Brno, fond G 12 Cerroniho sbírka 1200–1845, inv. č. 362: Německá kronika Nového Jičína z let 1607–1647.

AS6: Moravský zemský archiv Brno, fond G 21 Staré tisky 1469–1860, sign. III/160 (XIV n 236): Letopisecké přípisky anonymního autora k brněnskému exempláři Lupáčova Kalendáře historického z roku 1584.

AS7: Sächsische Landesbibliothek – Staats- und Universitätsbibliothek Dresden, 1 276: Ursinus, J. F., Collectanea zur Geschichte der Stadt und des Landes Meißen, Vol. 2.

AS8: Staatsarchiv Schaffhausen, Chroniken b5: Thurn, H., Chronik, 17. Jahrhundert.

AS9: Staatsarchiv Schaffhausen, Chroniken B6: Wepfer, G. M., Chronik 3 Vols.

AS10: Stadtarchiv Naumburg, SA 37, V: Varia, insonderheit zur Witterungs-Geschichte, vol. V (1601–1744).

AS11: Státní okresní archiv Louny, fond AM Louny – kroniky, sign. Ch1: Chronica civitatis Launensis in Boemia Auctore Paulo Mikssowicz servo consular.

AS12: Státní okresní archiv Teplice, fond AM Krupka, dodatky inv. č. 1: Knott, R., Michel Stüelers Gedenkbuch.

AS13: Státní okresní archiv Vsetín, fond AM Vsetín, sign. 129a: Památky rozličné událostí z let 1135–1838.

AS14: Stiftsarchiv Einsiedeln, mf 25: Brunschwiler, P., Diarium Fischingense 1616–1654.

AS15: Strahovská knihovna Praha, sign. AO.II.40: Letopisy Václava Arkadia připsané k druhému vydání Veleslavínova Kalendáře historického.



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
