# Peer review of "Weather and climate and their human impacts and responses during the Thirty Years' War in Central Europe"

_Climate of the Past, 2023_

## Referee Comment (RC2)

[referee-annotated manuscript omitted]

---

## Author Comment (AC1)

**Major comments:**

I find the article "Weather and climate and their human impacts and responses during the Thirty Years' War in Central Europe" by Rudolf Brázdil et al. highly interesting and generally well-written and of interest to a broad readership from several academic disciplines. However, the article is unfortunately not citing, or referring to, much of the newer key works in the climate change–human history nexus that I strongly would recommend the author to include during the revision phase of the article. Besides from that, I only have minor comments as listed below.

RESPONSE: We would like to thank the referee 1 for critical comments to the article which we are trying to respond below.

**Minor comments:**

Title: The title is a bit too long and complicated. Could the authors please consider to revise and shortened it?

RESPONSE: Thanks for this comment. We understand, that the referee may consider the title as long, but we would like to preserve the recent title which expresses clearly key topics related to the whole content of the article. Other order of words in the title (e.g., The Thirty Years' War in Central Europe: climate, weather, human impacts and response) is probably not an expected solution.

Lines 15, 50: The expression "17th century reformation" is not perfect as the reformation was mainly a 16th century phenomena.

RESPONSE: Accepted, the expression "17th century reformation" was deleted from sentences in both lines.

Lines 17–19, 676 and other places: I think it is wrong to talk about "deterioration" here as the late 16th century seems to have been even colder and even more adverse for agriculture in Central Europe. This is extensively discussed in, among other works, Pfister and Wanner (2021) cited by the authors.

RESPONSE: Thanks for this comment. We understand this traditional view to use this term for the late 16th century, but we believe that the expression "climate deterioration" is also acceptable talking about "a decrease in solar activity towards the Maunder Minimum and increased volcanic activity."

Line 31: Besides Tucker (2012), please also cite:
Lee, H. F., Zhang, D. D., Brecke, P., & Fei, J. (2013). Positive correlation between the North Atlantic Oscillation and violent conflicts in Europe. Climate Research, 56, 1–10.
Lee, H. F., Zhang, D. D., Brecke, P., & Pei, Q. (2015). Regional geographic factors mediate the climate–war relationship in Europe. British Journal of Interdisciplinary Studies, 2, 1–28.
Lee, H. F., Zhang, D. D., Brecke, P., & Pei, Q. (2019). Climate change, population pressure, and wars in European history. Asian Geographer, 36, 29–45.
Tol, R. S., & Wagner, S. (2010). Climate change and violent conflict in Europe over the last millennium. Climatic Change, 99, 65–79.
Zhang, D. D., Lee, H. F., Wang, C., Li, B., Pei, Q., Zhang, J., & An, Y. (2011). The causality analysis of climate change and large-scale human crisis. Proceedings. National Academy of Sciences. United States of America, 108, 17296–17301.

RESPONSE: Accepted, the new paragraph considering above papers was included into Introduction as follows: „For Europe, some papers tried to study causal connections of violent conflicts/wars with climate change (e.g., Tol and Wagner, 2010; Lee et al., 2015, 2019) or with the North Atlantic Oscillation (Lee et al., 2013). Similar, Zhang et al. (2011) investigated

the causality of large-scale human crisis with climate change. Their statistical approaches to analyze the relationship climate and society were criticised by Degroot et al. (2021, pp. 540–541) arguing that such studies "provide examples of the 'McNamara fallacy', in which unquantifiable data are either ignored or arbitrarily quantified to produce superficially impressive but potentially misleading results." Presenting a new interdisciplinary framework on the "history of climate and society," they concluded that the past climate change did not always lead to collapse or disaster and that populations survived or even thrived in the face of climatic pressure (see also White et al., 2023)."

Lines 32–33: Not sure how relevant this is here as it refers to weather events rather than climate.
RESPONSE: Thanks for this comment, which is not fully clear for us. In these lines there are references to studies that deal with the environmental aspects of the two World Wars and the American civil war. They don't focus only on weather events but also on climate.

Line 51: Maybe cite here:
Parker, G., 2006. The Thirty Years' War. Routledge, London.
Theibault, J., 1997. The demography of the Thirty Years War re-revisited: Günther Franz and his critics. Ger. Hist. 15, 1–21.
RESPONSE: Thanks for this comment. As for Parker (2006), the related paragraph starts with some introductory statements concerning of the Thirty Years' War, which are followed by six citations, including Parker (1997) which is the 2nd edition of the book (complemented by many other citations in Section 2). We believe that in this form it is acceptable, without adding additional citation.
As for Theibault (1997), it fits better to cite it at line 609: "Estimating the exact population of the Holy Roman Empire during that time is challenging (Theibault, 1997), but it is believed to have been around 16 to 18 million in approximately 1618, and decreased to 10 to 12 million by around 1650 (Wilson, 2009; Repgen, 2015; Gotthard, 2016)."

Line 55: This statement is not entirely true. Many studies have also considered decadal and inter-annual time-scales. See, for a review, Ljungqvist et al. (2021) listed below.
RESPONSE: Accepted and corrected as follows: "climatological patterns during this time have been analyzed not only on millennial or centennial scales (e.g., Glaser, 2008; Parker, 2008, 2013; Pfister et al., 2018; Pfister and Wanner, 2021), but also considered decadal or inter-annual scales (see Ljungqvist et al., 2021 for a review)."

Lines 199–201: The low skill of the Pauling et al. (2006) precipitation reconstruction prior to c. 1750 needs to be critically discussed and acknowledged here.
RESPONSE: Accepted. We are aware of the problem of uncertainty (the low skill), which applies to all used reconstructions. Therefore, it is difficult to discuss it in greater detail in this paper. We believe, that existing higher uncertainty of Pauling et al. (2006) data before 1750 is well explained by sentence added to point (iv) in Section 3.2: "Despite the fact that this precipitation dataset uses especially for the early 17th century a limited set of input proxy data (see Fig. 1 in Pauling et al., 2006) as well as rather out-of-date climate model for the gridded reconstruction (Ljungqvist et al., 2022), it is the only available spatial precipitation reconstruction with a seasonal resolution."

Line 201: Wrongly written. Should be: Self-calibrating Palmer Drought Severity Index (scPDSI).
RESPONSE: Accepted and corrected.

Around line 205: Consider to also here include the EuroMed2k (Luterbacher et al., 2016) reconstruction discussed later in the article (lines 570ff.).

RESPONSE: Thanks for this comment, but the JJA series by Luterbacher et al. (2016) is not used in results of the paper, but only for comparisons in Discussion, i.e., we believe that this paper should be not cited in Section 3.2.

Lines 209–210: I would strongly recommend to use the volcanic dataset of Toohey and Sigl (2017) instead of Crowley and Unterman (2013) as the later dataset is NOT state-of-the-art anymore and even includes dating errors. A revised article should use Toohey and Sigl (2017):

Toohey, M. and Sigl, M. (2017). Volcanic stratospheric sulfur injections and aerosol optical depth from 500 BCE to 1900 CE, Earth System Science Data 9: 809–831.

RESPONSE: Accepted. We used dataset of Toohey and Sigl (2017) and complemented this reference instead of Crowley and Unterman (2013). The last sentence of the second paragraph of Section 5.1.4 was changed as follows: "When considering the entire 17th century, a slightly stronger volcanic signal besides the 1640s was found at the beginning of the 1600s and in the mid-1690s."

Using data by Toohey and Sigl (2017), the new version of Fig. 8b was prepared – see below:

[Figure]

Lines 232 and other places: Ljungqvist et al. (2022) should be referred to, and discussed, concerning grain prices as it very extensively treats both the impacts from climate and from the Thirty Years' War on European grain prices. Reference:

Ljungqvist, F. C., Thejll, P., Christiansen, B., Seim, A., Hartl, C., and Esper, J.: The significance of climate variability on early modern European grain prices, Cliometrica, 16, 29–77, 2022.

RESPONSE: Accepted. Results of the paper by Ljungqvist et al. (2022) were included in Discussion in a new paragraph after Figure 14 as follows: "Esper et al. (2017) pointed out that European gran price series were spatially less coherent during the TYW (see also Figure 12 for Central Europe) when also temperature–grain prices correlations completely fell. Ljungqvist et al. (2022) in their extensive statistical analysis of temperature effects on grain prices in early modern Europe excluded the period of the TYW altogether. They argued that the grain prices were rather strongly influenced by "the disintegration of established market forces and regional decoupling of trade" and that the war and its effects "weakened the climate signal in the grain prices".

Line 344 Caption): Define "persistent frost" clearly.

RESPONSE: Accepted. We added related explanations to the first paragraph of Section 5.2 as follows: "As is characteristic for analysis of qualitative daily weather observations (e.g.,

Pfister et al., 1999; Brázdil et al., 2003, 2019; Domínguez-Castro et al., 2015; Harvey-Fishenden and Macdonald, 2021), Lenke (1960) calculated corresponding numbers of days according to weather phenomena observed and reported by Hermann IV as follows: frost day – any occurrence of frost during the day; persistent frost day – frost continuing the whole day; hot (very hot) day – any occurrence of heats during the day; precipitation day – the occurrence of rain, rain with snow, snowfall or hail/sleet during the day."

Line 529: Describe method calculating breakdates here.
RESPONSE: Accepted. At the end of the Section 4 the following explanation was added: "The method tests the possible occurrence of changes in the slope parameters of the linear regression models, which are gradually fitted to the grain price time series. In the first step, the F-test determines whether the series contains significant changes in the regression models. In the second step, the optimal number of changes (breakpoints) and the date of their occurrence are determined using the Bayesian Information Criterion, BIC (Bai and Perron, 2003)."
New reference:
Bai, J. and Perron, P.: Computation and analysis of multiple structural change models. J. Appl. Econometrics, 18, 1–22, https://doi.org/10.1002/jae.659, 2003.

Lines 637–639: I think it is an overstatement to claim that the destructive forces of the war was the MAIN cause of famine. Climate and war interacted in a destructive synergy. See, among other works, Slavin (2016) listed below.
RESPONSE: Accepted and corrected as follows: "The war shifted the balance between the determinants in Fig. 13. Climatic factors and wars interacted in a destructive synergy. Their effects devastated the available farmland, decimated livestock, burdened subjects with war taxes and tributes, made the populations more susceptible to disease, and led to a significant loss of the workforce through death, disease, and military duties, which likely exacerbated subsistence crises, food shortages, and famines (Outram, 2001; Slavin, 2016)."

**Missing references to important works (including overviews):**
Adamson, G. C., Nash, D. J., and Grab, S. W.: Quantifying and reducing researcher subjectivity in the generation of climate indices from documentary sources, Clim. Past, 18, 1071–1081, 2022.
RESPONSE: Accepted, this paper was cited at the end of the first paragraph of Sect. 5.2.1 as follows: "This discrepancy may be attributed to problems with the documentary data (missing monthly indices) and the different precipitation variability observed across various parts of Central Europe as well as in potential subjective generation of precipitation indices (see e.g. Adamson et al., 2022)."

Degroot, D., Anchukaitis, K., Bauch, M., Burnham, J., Carnegy, F., Cui, J., de Luna, K., Guzowski, P., Hambrecht, G., Huhtamaa, H., Izdebski, A., Kleemann, K., Moesswilde, E., Neupane, N., Newfield, T., Pei, Q., Xoplaki, E., and Zappia, N.: Towards a rigorous understanding of societal responses to climate change, Nature, 591, 539–550, 2021.
RESPONSE: Accepted, the citation of this paper was included in the new paragraph in Introduction (see above).

Haldon, J., Mordechai, L., Newfield, T. P., Chase, A. F., Izdebski, A., Guzowski, P., Labuhn, I., and Roberts, N.: History meets palaeoscience: Consilience and collaboration in studying past societal responses to environmental change, Proc. Natl. Acad. Sci. USA, 115, 3210–3218, 2018.

RESPONSE: Thanks for this comment, but this citation was not included being not really dealing with violent conflicts or wars.

Ljungqvist, F. C., Seim, A., and Huhtamaa, H.: Climate and society in European history, Wiley Interdiscip. Rev. Clim. Change, 12, e691, 2021.
RESPONSE: Accepted, this citation was added to as follows: "… climatological patterns during this time have mostly been analyzed not only on millennial or centennial scales (e.g., Glaser, 2008; Parker, 2008, 2013; Pfister et al., 2018; Pfister and Wanner, 2021; Wanner et al., 2022), but also considered decadal or inter-annual scales (see Ljungqvist et al., 2021 for a review)."

Slavin, P.: Climate and famines: A historical reassessment. Wiley Interdisciplin. Rev.: Clim. Change, 7, 433–447, 2016.
RESPONSE: This citation was added to the changed text in lines 637–639 – see above.

Wanner, H., Pfister, C., and Neukom, R.: The variable European Little Ice Age, Quat. Sci. Rev., 287, 107531, 2022.
RESPONSE: Accepted, this citation was included in the fourth paragraph of Introduction (see response to Ljungqvist et al., 2021 above).

White, S., Pei, Q., Kleemann, K., Dolák, L., Huhtamaa, H. and Camenisch, C. (2023) New perspectives on historical climatology. Wiley Interdisciplin. Rev.: Clim. Change, 14, e808.
RESPONSE: Accepted, this citation was included in the second and third paragraphs of Introduction.

---

## Author Comment (AC2)

**Response to comments by Neil Macdonals**

This is a well written and detailed paper exploring the impact of weather and climate on societies during the Thirty Years War in Central Europe. The paper provides a detailed analysis, which is sound and robust, in places some additional detail is required to improve clarity or expression, with an annotated copy of the manuscript attached to help the authors undertake these changes. I have also flagged in a couple of places sections where I felt the arguments presented warrented reflection.

In reading the paper to disentangle the human from the environmental factors I felt it would be beneficial to have a records or data from outside the area impacted by the TYW, as such I have added some comments based on English datasets for the comparable period, which help to show this. I hope this is helpful, and was actually added before I had read the final line of your conclusion which makes this exact point.

I have also suggested some additional references that may be of interest.

Neil Macdonald
University of Liverpool

RESPONSE: We would like to thank the referee Neil Macdonald for proposed corrections in the manuscript, which were completely accepted. As for further comments, we are trying to respond below referring to the original line/part of the referee comment directly in the manuscript:

The beginning of the manuscript: I appreciate outside your regional scope, but W.G Hoskins wrote an excellent paper on grain/climate for this period based on English data.
This is of interest for two reasons, the first is that it suggests harvests in England were good during this early period "the year 1620 saw the most abundant harvest within living memory". However, the 1630s-40s were poor with high prices, with wet weather key; the 1650s were better. This mirrors the records seen within this paper for C. Europe. It does though suggest that there would likely have been poor harvests in Europe under normal conditions, therefore the TYW exacerbated these impacts.
The second point is that it can be beneficial to use a country with limited impacts from this period, the English reformation occurs earlier and they have limited input into the TYW (recent evidence suggests 50,000 soldiers in Dutch (+other) forces may have been British), so could offer a good contrast, which would help distinguish the socio-economic from environmental drivers. [You make this comments in the final sentence I note having now finished the paper.]
Hoskins W.G. 1968. Harvest fluctuations and english economic history, 1620-1759, The Agricultural History Review, 15-31
A scanned copy can be found at https://www.bahs.org.uk/AGHR/ARTICLES/16n1a2.pdf
RESPONSE: Accepted, the following sentences comparing situation in England, Switzerland and Germany were added to the last paragraph in Section 6.2:
"For example, in England (with limited input of the TYW), harvests from 1626 to 1628 were excellent, while a sequence of deficient harvests appeared between 1646 and 1649 (Hoskins, 1968). In the neutral Swiss Confederation, albeit situated closer to the military conflict than England, spelt prices in Zürich peaked in 1622 and 1623 due to a harvest failure in 1621 and a coin debasement in 1622. Another peak in 1627 and 1628 resulted from a widespread harvest failure on the continent (see also Sect. 5.3). Prices peaked between 1632 and 1638, when war

raged several times near Switzerland, and the Confederation had to provide horses and grain (Schmidt, 2010). Prices in the 1640s were relatively low, except in 1642 (Schmidt, 2010; Studer, 2015). In Nuremberg (Germany), rye prices were high from 1621 to 1628 and again from 1632 to 1636, while those in the 1640s were unremarkable low up to 1648 (Bauernfeind, 1993; cf. Fig. 14). Probably, the spatial distance from the war seems to have mattered, apart from crop failures."

Line 22: "Remarkably below-mean values, centered around the 1630s, characterized precipitation and drought fluctuations." - this does not make sense
RESPONSE: Accepted, the sentence was deleted.

The first paragraph of Introduction: See some of the work by de Kraker for Low countries
RESPONSE: Thanks for this comment, but we did not find any his paper relevant directly to the topic to be included in Introduction. The period of the Thirty Years War was covered only in Hydrological Sciences Journal, 2006 (floods), Environment and History, 2013 (storminess), Hydrology and Earth System Sciences, 2015 (flooding in river mouths), and Water History, 2017 (the removal of ice on waterways), i.e. only with storminess and floods, outside of Central Europe.

The first paragraph after Fig. 3: Would it be possible to state the number of datapoints, or datasets that are contributing to the records at each point in time? This is often low with high uncertainties
RESPONSE: Accepted. We are aware of the problem of uncertainty, which applies to all used reconstructions. Therefore, it is difficult to discuss it in greater detail in this paper. We believe, that existing higher uncertainty of that data before 1750 is well explained by sentence added to point (iv) in Section 3.2: "Despite the fact that this precipitation dataset uses especially for the early 17th century a limited set of input proxy data (see Fig. 1 in Pauling et al., 2006) as well as rather out-of-date climate model for the gridded reconstruction (Ljungqvist et al., 2022), it is the only available spatial precipitation reconstruction with a seasonal resolution."

Line 291: state dataset used to calculate scPDSI
RESPONSE: Thanks for this comment. It is explained in point (v) in Section 3.2 as follows: "(v) Seasonal and annual self-calibrating Palmer Drought Severity Index (scPDSI; Palmer, 1965) for the Czech Lands (1501–2015 CE) (Brázdil et al., 2016), derived from Central European temperature and Czech precipitation reconstructions (Dobrovolný et al., 2010, 2015)." It means, that scPDSI values were calculated using quantitative temperature and precipitation reconstructions, described in the same section under points (i) and (iii).

Sect. 5.2, the first paragraph: how do you define hot days here? P>0mm?
Fig. 9: define permanent frost.
RESPONSE: Accepted. We added related explanations to the first paragraph of Section 5.2 as follows: "As is characteristic for analysis of qualitative daily weather observations (e.g., Pfister et al., 1999; Brázdil et al., 2003, 2019; Domínguez-Castro et al., 2015; Harvey-Fishenden and Macdonald, 2021), Lenke (1960) calculated corresponding numbers of days according to weather phenomena observed and reported by Hermann IV as follows: frost day – any occurrence of frost during the day; persistent frost day – frost continuing the whole day; hot (very hot) day – any occurrence of heats during the day; precipitation day – the occurrence of rain, rain with snow, snowfall or hail/sleet during the day."

Line 409: "May witnessed an over-reproduction of cockchafers, causing damage in Bohemia …" cockchafers explain??

RESPONSE: Accepted, we changed this sentence as follows: "May witnessed an over-reproduction of cockchafers, causing damage to fruit trees in Bohemia (AS11; Lisa, 2014)."

Sect. 5.4.1, the first paragraph: Is one of the key factors here that war also leads to loss of the young and often male parts of communities, which can result in reduced capacity to farm, this is well documented across France during the Napoleonic wars.

RESPONSE: Accepted. The following part of the manuscript was corrected as follows: "The obligation to house troops, which fell upon the townspeople wherever the soldiers happened to be, furthermore led to the draining of local resources, scarcity of grain and food, hunger, poverty, the risk of diseases spreading, and general hardship for all involved. The war also contributed to the loss of able-bodied young males, which resulted in a reduced capacity to farm. Moreover, the destruction of dwellings and tools and the loss of cattle need to be mentioned (Asch, 1997; Wilson, 2009; Münkler, 2017; Stoffel et al., 2022)."

Line 499: "Documentary sources indicate that the overpopulation of mice …" Provide source to support

RESPONSE: This introductory sentence of this paragraph expresses generally occurrence of mice and their damaging effects to express effects of mice to grain crops. Following sentences cite already corresponding sources where such information appears. We believe that any other sources are not needed here.

Line 626: General pattern in England during this period, based on sources such as Broadberry et al 2015.

RESPONSE: Thanks for this comment, but we believe that description of general patterns in England is out of scope of this paper.

Line 631: Hoskins noted above argues this in England post 1650s

RESPONSE: Thanks for this comment, but looking on the rather general statements from Pfister and Wanner (2021) in lines 626–632 we do not see as suitable and consistent with points (i)–(iv) to add here in point (v) this local citation by Hoskins (1968) paper.

Lines 637-638: "Subsistence crises and famines were primarily caused by the war and its effects, devastating farmland, decimating livestock …" I am uncomfortable with this - an argument could easily be presented that the crises and famines would have happened anyway because of the weather, but are likely exacerbated by the war... similarly plagues etc,

RESPONSE: Accepted and corrected as follows: "The war shifted the balance between the determinants in Fig. 13. Climatic factors and wars interacted in a destructive synergy. Their effects devastated the available farmland, decimated livestock, burdened subjects with war taxes and tributes, made the populations more susceptible to disease, and led to a significant loss of the workforce through death, disease, and military duties, which likely exacerbated subsistence crises, food shortages, and famines (Outram, 2001; Slavin, 2016)."

Figure 14: explain what the lines mean here

RESPONSE: Accepted, following sentence was added to figure caption: "Horizontal lines indicate mean prices for three different time intervals (1500–1567, 1568–1629, 1630–1670)."

Lines 691-694: "The large armies, numbering in the thousands, significantly increased the demand for grain." Would be good to document this more earlier within the manuscript if any evidence remains.

RESPONSE: Thanks for this comment. This was mentioned, albeit a bit more indirectly, in lines 459–466. Here we discuss that all powers, except for the Dutch, had insufficient means to support their troops adequately with logistics, including food and grain, which was part of the reason that led some soldiers to undisciplined behaviour including looting.

"It is important to investigate the significance of climatic variations and extreme events using the example of territories that were spared from war, although it should be noted that epidemics did not stop at territorial borders. This study serves as a starting point for such investigations." Would be good to document this more earlier within the manuscript if any evidence remains.

RESPONSE: Thanks for this comment. We agree fully with the referee, but in this extremely broad and complex topic it is extremely difficult to bring evidence for every aspect mentioned in our paper. From this reason the last sentence of the paper was complemented as: "This study serves as a starting point for such future investigations."

References: Recommend to cite:
Adamson, G. C., Nash, D. J., and Grab, S. W.: 2022. Quantifying and reducing researcher subjectivity in the generation of climate indices from documentary sources, Clim. Past, 18, 1071–1081
Harvey-Fishenden, A., & Macdonald, N. (2021). Evaluating the utility of qualitative personal diaries in precipitation reconstruction in the eighteenth and nineteenth centuries. Climate of the Past, 17(1), 133-149. doi:10.5194/cp-17-133-2021

RESPONSE: Accepted. The first paper was cited at the end of the first paragraph of Sect. 5.2.1 as follows: "This discrepancy may be attributed to problems with the documentary data (missing monthly indices) and the different precipitation variability observed across various parts of Central Europe as well as in potential subjective generation of precipitation indices (see e.g. Adamson et al., 2022)." The second paper was newly cited in the first paragraph in Section 5.2 (see point above).